# Near-infrared-laser-navigated dancing bubble within water via a thermally conductive interface

Man Hu [1,3] ✉, Feng Wang [1,3], Li Chen[1], Peng Huo[1], Yuqi Li[1], Xi Gu [1], Kai Leong Chong[2] & Daosheng Deng [1] ✉

Precise manipulation of droplets or bubbles hosts a broad range of applications for microfluidic devices, drug delivery, and soft robotics. Generally the existing approaches via passively designing structured surfaces or actively applying external stimuli, inherently confine their motions within the planar or curved geometry at a slow speed. Consequently the realization of 3D manipulation, such as of the underwater bubbles, remains challenging. Here, during the near-infrared-laser impacting on water, by simply introducing a thermally conductive interface, we unexpectedly observe a spontaneously bouncing bubble with hundreds-of-micrometer diameter at tens-of-Hertz frequency. The unique formation of temperature inversion layer in our system generates the depth-dependent thermal Marangoni force responsible for the bouncing behavior. Both the scaling analysis and numerical simulation agree with observations quantitatively. Furthermore, by controlling the navigation speed of the laser beam, the bubble not only shows excellent steerability with velocity up to 40 mm/s, but also exhibits distinctive behaviors from bouncing to dancing within water. We demonstrate the potential applications by steering the bubble within water to specifically interact with tiny objects, shedding light on the fabrication of bubble-based compositions in materials science and contamination removal in water treatment.

Precise manipulation and particularly the directional transport of either droplets or bubbles are essential for technological advancement in microfluidics and Lab-on-a-Chip[1], materials science, thermal management[2], autonomous fluidic machines[3], and soft robotics[4]. Existing strategies mainly rely on the bio-inspired design of the structured surfaces through topographic pattern, wettability gradient, surface charge, or capillary ratchet[5,6], in order to realize the continuous horizontal translation. Nevertheless, the structured surfaces unavoidably confine the motion within the planar or curved geometry. Other possible approaches utilize the external stimuli to actively drive the motion of droplets or bubbles, including magnetic, electric, acoustic

or light fields[7-11], and their dynamics is correspondingly restricted by the applied specific forces. In the well-chosen binary liquids (propylene glycol and water), the vapor-mediated sensing and motility leads to the long-range interaction of droplets, but suffers from the limited strength and slow moving speed[3,12].

Recently, the counterintuitive bouncing of droplets or bubbles along the vertical direction has been exploited, such as droplets bouncing on a vibrating fluid bath[13-15], spontaneous droplet trampolining on rigid superhydrophobic surfaces in a low-pressure environment[16], or the air bubble bouncing at a compound interface of air/oil/water[17]. Particularly, arising from the effect of physicochemical hydrodynamics in a

[1]Department of Aeronautics and Astronautics, Fudan University, Shanghai 200433, China. [2]Shanghai Key Laboratory of Mechanics in Energy Engineering, Shanghai Institute of Applied Mathematics and Mechanics, School of Mechanics and Engineering Science, Shanghai University, Shanghai 200072, China. [3]These authors contributed equally: Man Hu, Feng Wang. ✉e-mail: human@fudan.edu.cn; dsdeng@fudan.edu.cn

binary liquid (ethanol and water)[18], oil droplet spontaneously displays the self-propelling behavior due to the gravitational stratification[19–21], and the plasmonic bubble shows the periodic bouncing response above the nanoparticle-decorated substrate[22,23]. However, it remains challenging to fulfill the flexible and multi-dimensional manipulations in a simple system remotely or noninvasively.

Here, by adopting a near-infrared laser irradiating into pure water through a transparent glass cover with a high thermal conductivity, we observe a vertical bouncing behavior of an optothermal bubble. The physical mechanism of the bouncing bubble is proposed by taking into account the formation of the temperature inversion layer (TIL) and the associated thermal Marangoni forces along the vertical direction. We perform the scaling analysis and numerical simulation, which agree with observations excellently. Moreover, being subject to the navigation of the laser beam, the resulted dancing bubble can be translated along the horizontal direction, showing a remarkable steerability with the speed up to 40 mm/s. Further, through the precise control of the specific interaction with the preexisting objects, we successfully demonstrate the unique bouncing and steerability features of this dancing bubble, consequently shedding light on the bubble-based compositions (such as bubble/droplet capsules or soft robots[4]) for fabrication in materials science, micro-reaction in chemical engineering, wastewater treatment in environmental science, and targeted drug delivery in bioengineering.

## Results

### Observation of the bouncing bubble

A near-infrared 980-nm laser impacts upon water by passing through a 250-μm-thick sapphire glass, which is optically transparent at 185–5000 nm (Fig. 1a). Due to the absorption coefficient of 0.45/cm at

980 nm wavelength for water, a vapor bubble is formed within the water at an elevated temperature, once the laser power exceeds the threshold value, $P \approx 15$ W (beam size $2r_1 \approx 1$ mm, laser intensity $\approx 1.9 \times 10^3$ W/cm²)[24] (see Methods, Supplementary Fig. 1). Remarkably, rather than the upward rising from the buoyancy force, the bubble exhibits the periodic bouncing, as shown in high-speed imaging at $t = 26$ s (Fig. 1a and Supplementary Movie 1).

During the bubble growth within 50 s, the bubble radius has a tendency to increase linearly with time (Fig. 1b and Supplementary Note 1). The remarkable observation of the continuous and periodic bouncing of the bubble endures for tens of seconds, as marked by the yellow regime, once its radius is satisfied with $R_{low} < R < R_{up}$. The position of the bubble center ($H_c$), which is characterized by the vertical distance between the bubble center and water/glass surface, oscillates rapidly with bouncing frequency about 17 Hz (Fig. 1c).

### Formation of temperature inversion layer

The thermal imaging during one period of bouncing bubble (Fig. 2a) shows the temperature profile does not monotonically decay away from the glass/water interface, as described by Beer-Lambert law[25–27]. By quantifying temperature distribution along the laser pathway (the green line in Fig. 2a), a TIL with thickness $\delta_{inv}$, within which temperature gradually increases to a peak temperature ($T_{peak}$), is clearly identified with thickness around 0.30 mm (Fig. 2b). The magnitude of the temperature gradient $dT/dZ$ within such inversion layer, reaches about 50 K/mm.

The existence of TIL is attributed to the cooling effect of the glass cover. As swept by the thermally-driven buoyancy flow during laser heating (Fig. 2c)[28], the thickness of viscous boundary layer is $\delta_v = 5L/Re^{1/2} = 0.40$ mm, and the thickness of thermal boundary layer

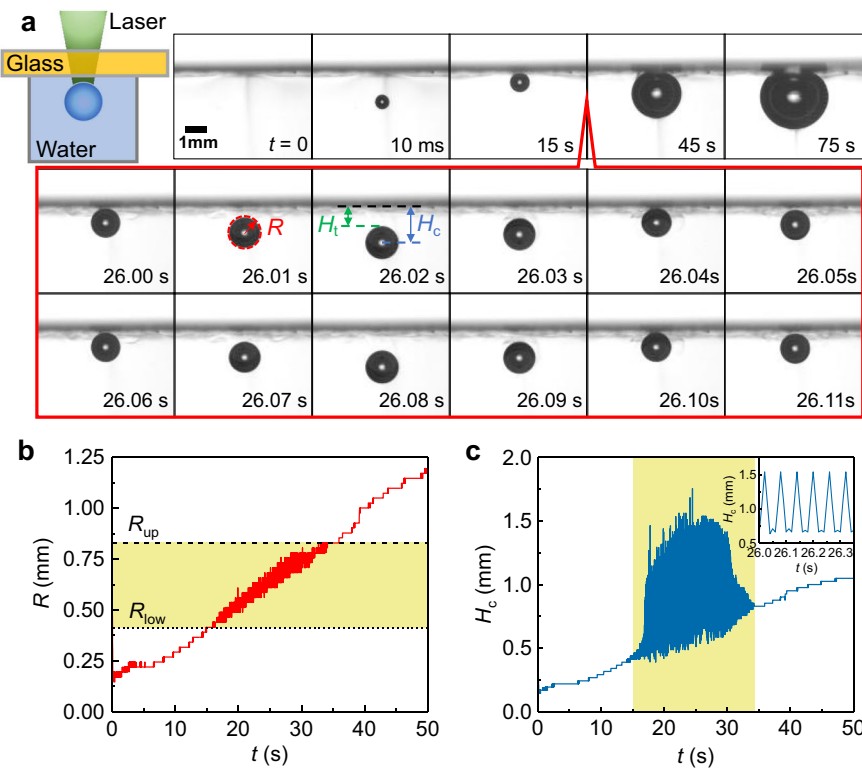

**Fig. 1 | Observation of a bouncing bubble within water. a** Sketch of the experimental setup, a continuous near-infrared laser (with 980-nm wavelength at $P = 15$ W) impacting on water through a superaerophobic, transparent and conductive glass. High-speed images for the produced bubble within water, and the two typical periods for the observed bouncing behavior presented in the red box ($t = 0$ s for the moment right before the bubble is visible; $H_t$ and $H_c$ for the topmost and the center of the bubble). **b** The growth of bubble radius $R$ with time $t$, and the bouncing regime highlighted in the yellow shade between a lower bound ($R_{low}$) and an upper bound ($R_{up}$). **c** The bouncing behavior characterized by the oscillation of the bubble center ($H_c$) with frequency about $f_b = 17$ Hz as shown by the inset. Source data are provided as a Source Data file.

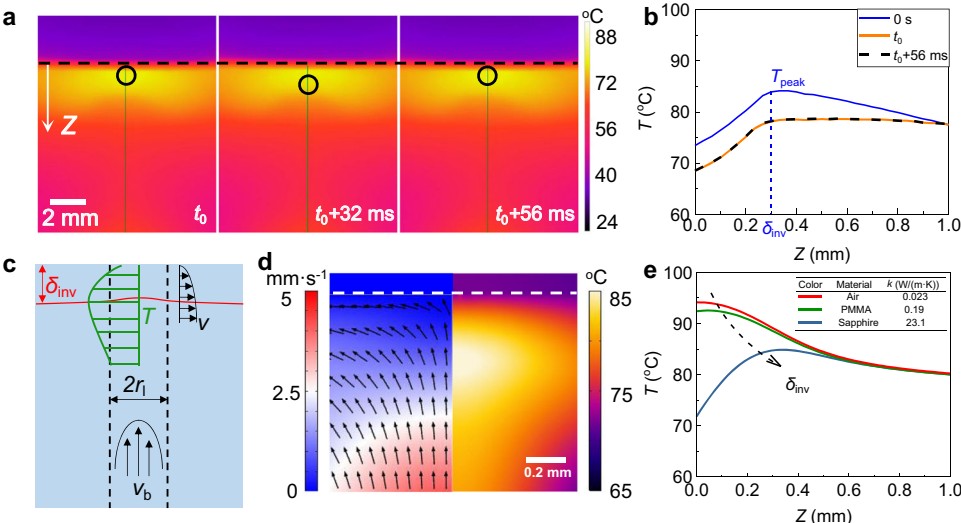

**Fig. 2 | Formation of TIL. a** Snapshot of the thermal imaging from side view during one period of bubble bouncing at $P = 15$ W. The black dash line for the glass/water interface, the circle for the bubble position, and the green line for the $Z$ direction ($t_0 = 18.48$ s). **b** Temperature profile along the $Z$ direction ($Z = 0$ refers to the glass/water interface), indicating a pronounced temperature peak $T_{peak}$ at

$\delta_{inv}$ about 0.3 mm and a strong temperature gradient $dT/dZ$ about 50 K/mm (Supplementary Fig. 4). **c** Sketch for the formation of TIL. **d** Simulation for the flow and temperature, confirming $T_{peak}$ for TIL. **e** $T_{peak}$ arising from the high thermal conductivity of the sapphire glass in simulation. Source data are provided as a Source Data file.

is $\delta_{th} = \delta_v / Pr^{1/3} = 0.33$ mm (Methods, Supplementary Note 5). Since sapphire has a higher thermal effusivity or conductivity, the cooling effect of cover becomes significant, resulting in a pronounced TIL with a lower temperature at the interface.

TIL is further confirmed by the simulation for the heat transfer based on the convection-diffusion model (Fig. 2d), and its thickness ($\delta_{inv}$) indeed is around hundreds of micrometers below the interface (Supplementary Methods). This $T_{peak}$ is directly caused by the high thermal conductivity of the sapphire glass, which is gradually diminished for a low thermal conductivity of PMMA or air (Fig. 2e and Supplementary Note 2).

Quantitatively, the thickness of TIL ($\delta_{inv}$) can be related with that of thermal boundary layer ($\delta_{th}$) as following,

$$\delta_{inv} = \xi \delta_{th}, \tag{1}$$

where $\xi$ ($0 < \xi < 1$) is a dimensionless number related with the thermal properties of the cover materials and water, i.e., the ratio of their thermal effusivity $e = \sqrt{kc_p\rho}$ ($k$, $c_p$, $\rho$ for the thermal conductivity, heat capacity, and density). As $\xi$ increases with $e$, in the limit of high thermal effusivly or thermal conductivity, the thickness of TIL nearly approaches that of the thermal boundary layer ($\delta_{inv} \approx \delta_{th}$) (Supplementary Note 6). Particularly, for sapphire with a high thermal conductivity, within TIL, the temperature gradient up to 50 K/mm (Supplementary Note 3) might be responsible for the bouncing behavior.

**Mechanism of the bubble bouncing**

In order to reveal the underlying physical mechanism of the observed bubble bouncing behavior, we take into account the identified TIL by carefully investigating thermal Marangoni force under the assumption of a constant temperature gradient either within or outside TIL. Indeed, the direction of the thermal Marangoni force can be switched from upward to downward direction[29], depending on the location and motion status of the bubble (Fig. 3a, c).

When the bubble moves in liquid (Fig. 3a), both the buoyancy force $F_b = 4\pi\rho g R^3/3 \propto R^3$ and the upward Marangoni force $F_m^+ = \Delta\gamma \cdot R = \gamma_{th}^+ \cdot 2\pi R^2 \propto R^2$ ($\gamma_{th}^{\pm} = \frac{d\gamma}{dT}\frac{dT^{\pm}}{dz}$ for the gradient of surface tension[30]) serve as the restoring force, bringing bubble to the solid wall as

expected. When the bubble rests near the cover (Fig. 3c), the buoyancy force $F_b$ is the same, and the upward Marangoni force $F_m^+ = \gamma_{th}^+ \cdot \pi R \cdot \max(0, 2R - \delta_{inv})$ depends on the height outside TIL. However, the downward Marangoni force $F_m^- = \gamma_{th}^- \cdot \pi R \cdot \min(2R, \delta_{inv})$ becomes significant, enabling the possibility to push bubble downward away from the solid wall. From this scaling analysis, the comparison of the magnitude of these forces is shown in Fig. 3b, d.

Regarding this bouncing behaviour, the upper bound radius ($R_{up}$) can be evaluated by balancing buoyancy force ($F_b$) with the downward Marangoni force ($F_m^-$),

$$R_{up}^2 = \frac{3\gamma_{th}^-}{4\rho g}\delta_{inv}, \tag{2}$$

where the gradient of surface tension $\gamma_{th}^- \propto \Delta T \propto P$ (Supplementary Note 4), and the thickness of TIL $\delta_{inv} \propto P^{-1/4}$ (Supplementary Note 6), then $R_{up} \propto P^{3/8}$.

To evaluate the lower bound radius ($R_{low}$), the bubble bouncing height ($H_t$) needs to be clearly observable in experiments, and a critical bouncing height is introduced $H_{t,cr} \propto 0.1R_{low}$ by eliminating fluctuations of bubble radius. From energy perspective, the work done by the downward Marangoni force ($W_m^- \propto \gamma_{th}^- R^2 \delta_{inv}$) is mainly converted into the kinetic energy of bubble associated with an initial velocity ($v_{b,0}$), $E_k \propto \rho R^3 v_{b,0}^2$. Subsequently during the downward bouncing, due to the upward Marangoni force ($F_m^+ \propto \gamma_{th}^+ R^2$), the bouncing height or the travel distance can be evaluated from energy conservation, $E_k = W_m^+ = F_m^+ H_{t,cr}$, thus $H_{t,cr} \propto \gamma_{th}^- \delta_{inv}/\gamma_{th}^+ \propto P^{-1/4}$ (Supplementary Note 7).

The following scaling relation between bubble radius $R$ and laser power $P$ for the bouncing criteria is obtained,

$$\begin{cases} R_{up} \propto P^{3/8} \\ R_{low} \propto P^{-1/4} \end{cases} \tag{3}$$

which agrees well with the experiments (Fig. 3e). Once $R > R_{up}$, the dominant upward buoyancy prohibits the downward motion of bubble, and bouncing is ceased and bubble floats within water.

Additionally, the frequency of bouncing can also be characterized by the scaling analysis (Fig. 3f). For small bubble ($R < R_{cr}$ in Fig. 3b), the upward thermal Marangoni force $F_m^+$ dominates the bubble motion,

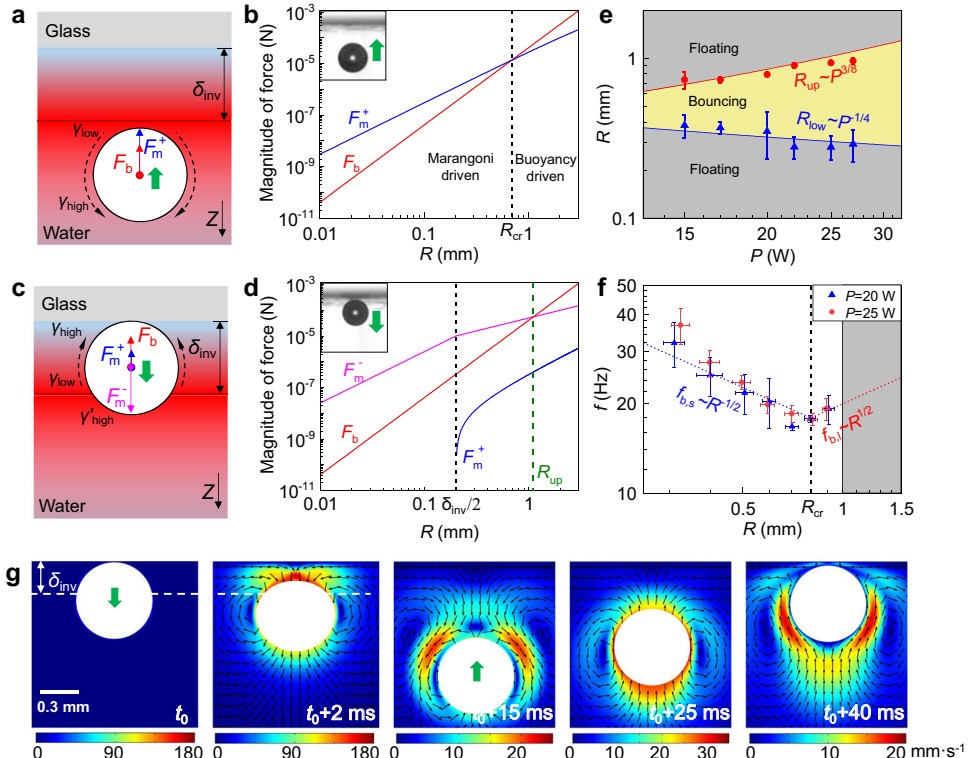

**Fig. 3 | Mechanism of the bubble bouncing. a–d** The relevant driving force responsible for the bouncing at $P = 20$ W. Sketch of the competing buoyancy force and the thermal Marangoni force, and their magnitudes dependent on bubble radius, when bubble moves in liquid (**a**, **b**) and rests near the wall (**c**, **d**), respectively. **e** The lower ($R_{low}$) and upper bound ($R_{up}$) for the bouncing behavior dependent on the laser power ($P$), the lines for theory, and dots for the experiments. **f** Bouncing frequency dependent on the laser power and bubble radius, lines for the theory, and dots for the experiments. **g** The numerical simulation conforming the bouncing bubble and revealing the related velocity field. The white dash line for the location of TIL. The error bars of the data in (**e**–**f**) denote the standard deviation of at least three measurements. Source data are provided as a Source Data file.

while for the large bubble, the buoyancy force becomes critical. Then the scaling of bouncing frequency dependent on the radius is identified,

$$f \propto \begin{cases} (F_m^+/\rho R^3)/\upsilon_{b,0} \propto R^{-1/2}, & (R < R_{cr}) \\ (F_b/\rho R^3)/\upsilon_{b,0} \propto R^{1/2}, & (R > R_{cr}) \end{cases} \quad (4)$$

By considering the interplay of buoyancy and Marangoni forces, the numerical simulation confirms the bouncing bubble subjected to a given TIL (Fig. 3g, Methods). The flow fields reveal the reverse flow or the switch of Marangoni flow direction when the bubble is outside TIL (Fig. 3g at $t = t_0 + 15$ ms).

**Bubble dancing with horizontal translation**

After elucidating the underlying physics of the bouncing bubble along the vertical direction, we introduce another degree of freedom, i.e., the translation movement with the laser beam (Fig. 4a), to further demonstrate the intriguing trajectory of the bubble under various horizontal speeds (Fig. 4b). At a lower speed of translation along the $x$ direction ($\upsilon_l = 1$ mm/s, Supplementary Movie 2; and 3 mm/s Supplementary Movie 3), the bubble not only presents the excellent steerability with the laser beam, but also preserves the bouncing behavior along the $z$ vertical direction, hence achieving the dancing motion of underwater bubble. This steerability is driven by the thermal Marangoni force along the $x$ direction to push bubble toward the hotter area of laser beam, since the thermal gradient ($dT/dx$) arises from the local heating in the course of the translation.

However, at a higher speed ($\upsilon_l = 5$ mm/s, Supplementary Movie 4), the bubble still can be translated by the laser beam, while the bouncing

behavior has been suppressed completely. If $\upsilon_l$ is further increased, then the bubble has insufficient time to respond and follow. The horizontal translation of the bubble is driven by the thermal Marangoni force along the $x$ direction,

$$F_m^x = \gamma_{th} \cdot 2\pi R^2 = \frac{d\gamma}{dT} \cdot \frac{dT}{dx} \cdot 2\pi R^2, \quad (5)$$

Here the temperature gradient ($dT/dx$) is caused by the local dwelling timescale ($\tau_d$) during the translation of laser beam, which is simply related with the translation speed ($\tau_d \sim 2r_l/\upsilon_l$). This translation driven force is decreased with $\upsilon_l$, while the Stokes viscous drag force is raised with $\upsilon_l$ ($F_v = -12\pi\mu R\upsilon_l$[31]). Due to the superaerophobic interface without adhesion, by balancing $F_m^x \sim F_v$[32,33], the obtained maximum translation speed is consistent with the experimental observation, which is about 40 mm/s (40 length of body per second for bubble diameter around 1 mm)[34] (Supplementary Movie 5, Supplementary Note 8).

The preserved bouncing behavior subjected to the translation is viewed from the necessary condition of temperature gradient $dT/dZ$ along the $z$ direction within TIL. Besides the dwelling timescale $\tau_d$, $\tau_c$ is the characteristic timescale for the heat transfer (defined in Supplementary Fig. 4). For the short time period ($\tau_d < \tau_c$), the elevated temperature of water ($\Delta T$) increases with time ($\Delta T \propto P\tau_d$, Supplementary Note 4), while for the long time period ($\tau_d > \tau_c$), the elevated temperature of water reaches a steady state ($\Delta T \propto P\tau_c$).

By substituting the scaling relation of $\gamma_{th}^- \propto \Delta T$ into the previous analysis (Eq. (2)), the upper bound radius for dancing bubble can be obtained, $R_{up}^2 = 3\gamma_{th}^-\delta_{inv}/4\rho g \propto P^{3/4} \min(\tau_d, \tau_c)$. Then, the following

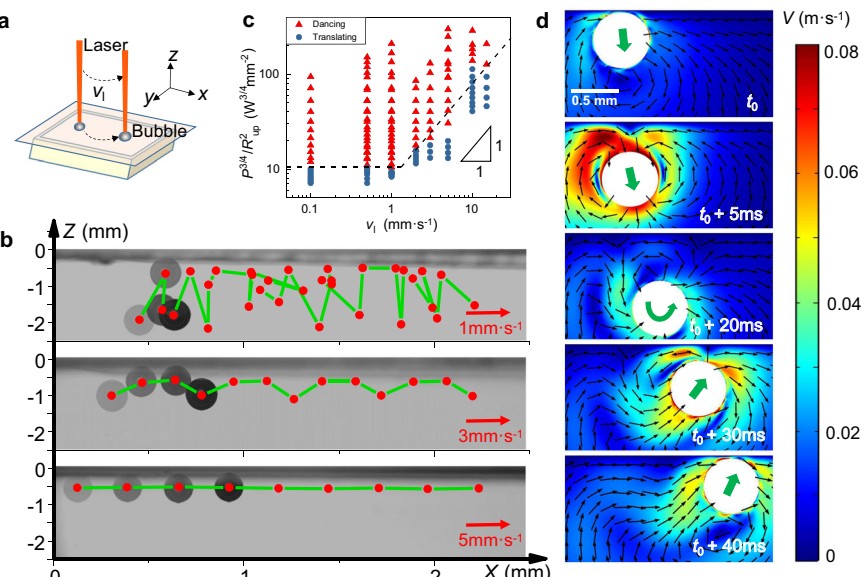

**Fig. 4 | Bubble dancing guided by the translation of laser beam. a** Sketch for the bubble following after the laser beam. **b** Images of bubble moving trajectory with translation speed of 1, 3, and 5 mm/s. **c** Phase diagram of bubble dancing dependent on the speed of the laser translation. **d** Simulation of the bubble dancing. Source data are provided as a Source Data file.

scaling is obtained,

$$P^{3/4}/R_{up}^2 \propto [\min(\tau_d, \tau_c)]^{-1} \sim \begin{cases} \tau_c^{-1} \sim \text{const.}, & (\tau_d > \tau_c, v_l < 2r_l/\tau_c) \\ \tau_d^{-1} \sim v_l, & (\tau_d < \tau_c, v_l > 2r_l/\tau_c) \end{cases} \quad (6)$$

which agrees excellently with experiments (Fig. 4c).

This dancing bubble is further validated by numerical simulation when the bubble is subjected to an additional horizontal temperature gradient (Fig. 4d, Methods). Although the flow fields are more complicated by the interplay of Marangoni, buoyancy and viscous forces, the bouncing behavior along the vertical direction is still reproduced in the course of horizontal translation.

## Utilizing the features of the dancing bubble

By taking advantage of the unique bouncing and steerability features of this dancing bubble, through the precise control of the specific interaction with the preexisting objects including walls, bubbles, droplets or nanoparticles, we successfully demonstrate its versatile capability to leap over a wall, coalesce with bubbles, form the core (bubble)-shell (droplet) structure, and collect particles within water.

As shown in Fig. 5a (Supplementary Movie 6), this spontaneous dancing behavior together with the horizontal motion under navigation of the laser allows the attainment of complex movement in three dimensions, as indicated by its prescribed well-defined trajectory of the cross shape. Generally, a bubble is easily to be pinned at the surface due to defects or microstructures at surface, but the dancing bubble might circumvent this detrimental effect of trapping. For example, by tuning the relevant physical parameters in simulation, such as bouncing frequency and the translation velocity, the dancing bubble can indeed leap over a wall (Fig. 5b, Supplementary Movie 7 and Supplementary Methods).

For the bubble coalescence (Fig. 5c), the dancing bubble *i* shows an excellent steerability with the laser beam along the *y* translational direction at the speed of 5 mm/s, encounters the preexisting bubble *ii* and *iii* along the pathway at 0.8 s, resulting in the sequential capture and fast coalescence at 0.9 and 1.0 s. Then the combined large bubble *i + ii + iii* continues its motion as guided by the laser (Supplementary Movie 8).

A core-shell structure of bubble-droplet composite is produced by steering the dancing bubble into a droplet (Fig. 5d). A 2.8-mm-diameter hexane droplet is dyed into orange color with the aid of organosulfur (tetrathiafulvalene) for better imaging, and is closely attached on the transparent solid glass (Supplementary Movie 9). By following after the laser beam with the translation speed of 1 mm/s along the *x* direction, a 0.6-mm-diameter dancing bubble is accurately guided into the droplet, resulting in an encapsulation of core-shell structure composed by bubble and droplet. This precise manipulation and remote control of bubbles facilitate the sophisticated bubble-droplet structures, providing practical strategies for the relevant applications in chemical engineering and biomedicine industry[35].

Additionally, in an aqueous solution by dispersing particles (polystyrene (PS), 80 μm diameter) into the deionized water, after the bubble sweeps through water under the navigation of the laser beam, the PS particles are adsorbed by the bubble (Fig. 5e and Supplementary Movie 10), and the trapping process shows excellent self-assembly and self-arrangement characteristics. Due to its downward and upward jumping during the horizontal movement, the dancing bubble certainly covers a larger volume of water and efficiently collects the particles through this manner of the deep surface cleaning. Hence the contaminations of particles are removed to purify water, in a way similar to self-propelled micromotors for water purification[36].

## Discussion

Conventionally, by utilizing the applied external stimuli (such as magnetic, electric, acoustic, and optical fields), the dynamics of droplets or bubbles can be accordingly controlled and their motions are subsequently directed[7–11]. For example, magnetic and electric actuation require magneto-responsive and dielectric surfaces, but the need to manufacture patterning electrodes on substrate or embedding magnetic particles into a soft matrix increases the complexity of the overall operation[7,8]. Optical and acoustic tweezers are powerful tools for the non-contact manipulation of bubbles[37,38]. For optical tweezers, the high degree of flexibility and fine spatial resolution is offered, but the achievable trapping force is quite weak on the order of pico-newtons, which can only be dominant at the micro-level[39]. Acoustic-based traps, which resemble optical traps, can accommodate larger samples with the size on the order of millimeters[40]. Although the

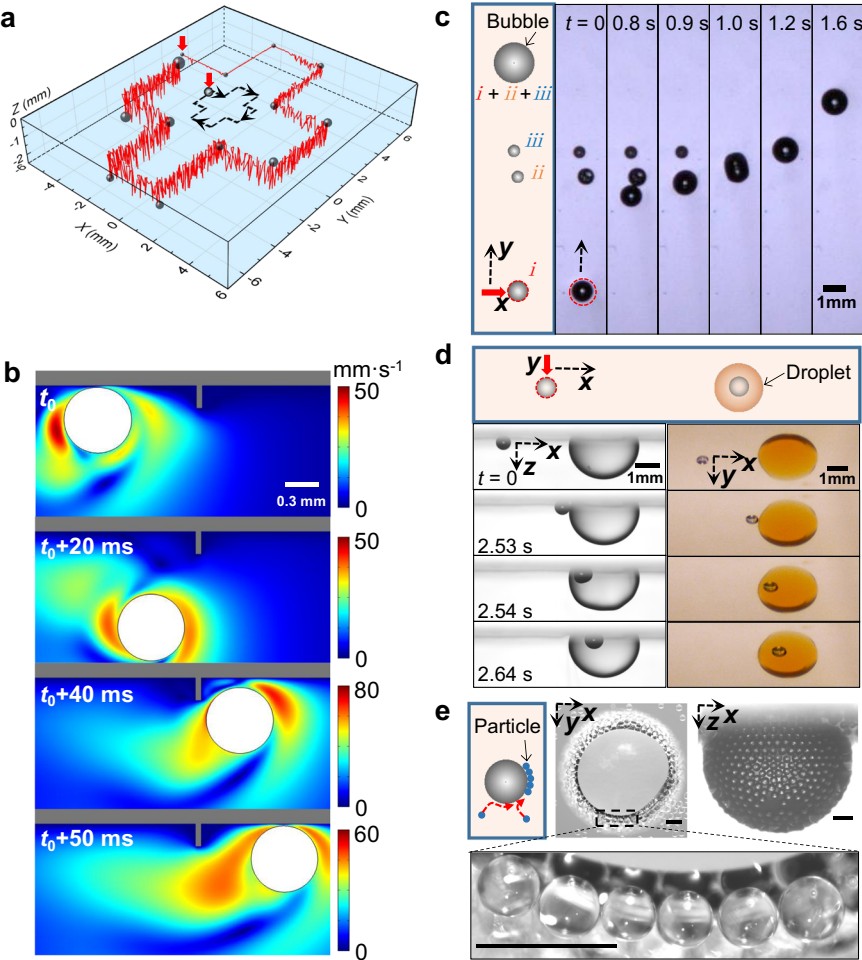

**Fig. 5 | Utilizing the features of the dancing bubble. a** 3D manipulation of the dancing bubble, as shown by its trajectory with the translation speed of 2 mm/s, the red arrow for the initial position. **b** Simulation for the dancing bubble to leap over a wall. **c** The coalescence of bubbles (*ii* and *iii*) with the dancing bubble (*i*) along its pathway under a moving speed of 5 mm/s. **d** The encapsulation of the dancing bubble by a dyed hexane droplet from the side and top view, producing a core-shell structure of bubble-droplet composite. **e** Particle collection unto the surface of the dancing bubble during its sweeping through the water, top and side view for PS beads (diameter of 80 μm) collected by the bubble. Scale bar for 200 μm. Source data are provided as a Source Data file.

hybrid trap by combing the two modalities has been implemented, more compact and versatile manipulation designs for multiple functionalities are needed to be further explored[41].

In contrast, the light-induced Marangoni effect, a natural way to convert light into thermal energy to induce Marangoni flows, can attract bubble or droplets with μm to mm size at a relatively high speed over distances significantly larger than the object size. Although diverse examples of bubble motion by applying the photothermal Marangoni effect have been demonstrated, its motion is restricted in 2D manipulation, and its performance is hindered by the strong adhesion of the surface-attaching bubble[32,42,43]. Attempt for 3D trapping and manipulation of microbubble in the bulk liquid relies on the necessary requirement for the fiber immersed into the liquid, impeding the noninvasive feature inherent with the optical approach.

We note that in the recent work in ref. 23, bouncing plasmonic bubble has been demonstrated in a binary liquid consisting of water and ethanol, and the competition between the solutal and thermal Marangoni forces is identified as the origin of the periodic bouncing. However, with only pure water as the host fluid here, both the bidirectional thermal Marangoni force and buoyancy force are exploited carefully to achieve bouncing. Also, the bouncing frequency and the size of bubble are different in orders of magnitude for both works (several kHz for micrometer bubble in ref. 23 versus several Hz for millimeter bubble in this work).

In order to produce the bouncing bubble, naturally the laser penetration depth in water should be large enough to produce spherical bubble and strong vertical temperature gradient within water. Generally, the glass on top of water meets three specific requirements of being optically transparent, thermally conductive and hydrodynamically superaerophobic. Firstly, the glass needs to be transparent at the near infrared wavelength, allowing 980-nm laser to pass through without strong attenuation of intensity. Otherwise, the strong attenuation of light intensity in glass, such as the far infrared wavelength of 10.6-μm-wavelength laser[44], hinders the bubble production in water. Secondly, the high thermal conductivity enables the fast heat transfer from water to the glass, forming a local temperature peak within TIL to facilitate the periodic bouncing. Thirdly, superaerophobic property of glass naturally prevents the bubble stick or adhesion to the glass, enabling both bouncing and rapid translating.

## Methods
### Experimental setup
The experimental setup is shown in Supplementary Fig. 1. Deionized (DI) water was contained in a quartz glass cuvette (20 × 20 × 10 mm) covered with a glass slide (sapphire, which needs to be optically transparent for the irradiating laser). A laser source of 980-nm wavelength was slightly focused on the liquid sample transmitted through the cover glass from the top side. Bubble dynamics was illuminated

from side and below respectively with two xenon lamps and imaged onto two synchronized fast-speed cameras (Phantom V611 and C110). An infrared thermal camera (FLIR A6750sc) was used to acquire the temperature evolution of local water during the laser irradiating.

In our experiments, the laser beam output from a fiber-coupled diode laser at 980-nm wavelength is first collimated by a plano-convex lens to a beam with diameter of about 2.2 mm. Then, the collimated laser beam is slightly focused by a focusing lens (L1) to a spot with diameter of about 1 mm. We use both the 2.2 mm collimated laser beam and the 1 mm slightly focused laser beam to generate the optothermal bubble, and observe the bouncing behavior for both laser beam, although the bubble is formed at the different threshold laser intensity.

An infrared thermal camera (FLIR A6750sc) was used to acquire the temperature evolution of liquid during laser irradiating. For the top view temperature detection, the sapphire glass with high transmissivity @185–5000 nm was adopted as the cover glass since infrared light can optically penetrate the sapphire glass. Since the side wall of the quartz cuvette affects the detection of infrared intensity signal emitted from the target liquid surface, the measured temperature values from the side view by the thermal camera was required to be calibrated. For the calibration, a thermocouple was exactly located at the same position of laser focus spot, closer to the quartz wall (about several millimeter). By using the thermal camera and the thermocouple to simultaneously measure the temperature evolution of a cooling process for a glycerol in the cuvette within the temperature range from 140 °C to 30 °C (the same emissivity with that of water, $\varepsilon = 0.96$), the elevated temperature of the irradiated liquid in the plane of laser transmission was quantitatively calibrated.

### Thermal boundary layer and temperature inversion layer

The thermal boundary layer ($\delta_{th}$) is formed by the laser-induced buoyancy flow sweeping the cover surface, which can be described by the hydrodynamic solution by Blasius[28]. The thickness of thermal boundary layer $\delta_{th} = \delta_v/Pr^{1/3}$ is related with the velocity of laser buoyancy flow and the properties of liquid, where Prandtl number $Pr = c_p\mu/k = 1.76$ for water at 373 K ($c_p$, $\mu$, $k$ for the heat capacity, dynamic viscosity, and thermal conductivity, respectively). Within the thermal boundary layer, the conductive cooling effect of cover is dominant, while outside the thermal boundary layer, the cooling effect is negligible. TIL ($\delta_{inv}$) represents the turning point of temperature distribution along $z$ direction, which is associated with the competition between laser heating and conductive cooling of cover. Thus, the thickness of TIL ($\delta_{inv}$) is dependent on thermal properties of liquid and solid cover. Derivation of relation between $\delta_{th}$ and $\delta_{inv}$ is provided in Supplementary Note 6.

### Numerical simulation for bubble bouncing and dancing

For bubble bouncing, the Navier–Stokes equation is solved in axisymmetric cylindrical coordinate ($r, \theta, z$), involving gravity term. For bubble dancing, the Navier–Stokes equation is solved in Cartesian coordinate ($x, y$), involving gravity term. And for both cases, bubble radius $R$ and thickness of TIL $\delta_{inv}$ are 0.3 mm. The temperature distribution along $z$ direction is simplified to a negative temperature gradient ($dT^-/dz = -50$ K/mm) within TIL ($0 < z < \delta_{inv}$) and a positive temperature gradient ($dT^+/dz = 2.5$ K/mm) outside TIL ($z > \delta_{inv}$). For bubble dancing, The temperature distribution along $x$ direction is simplified to a positive temperature gradient ($dT^+/dx = 10$ K/mm). The interface between water and gas is captured by phase field method. The surface tension is assumed as a linear function of temperature $\gamma(T) = \gamma_0 + \frac{d\gamma}{dT}(T - T_0)$, and the model is solved with COMSOL. Details for settings and simulated results are presented in Supplementary Methods.

## Data availability
The data that support the findings of this study are available from the corresponding authors upon request. Source data are provided with this paper.

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

## Acknowledgements

D.D. is grateful to H. Stone, Y. Fink, S. Johnson and J. Bush for the insightful and supportive comment on this work. M. H. acknowledges the National Natural Science Foundation of China (No. 11704077); K. L. C. acknowledges the Shanghai Science and Technology Program under project no. 19JC1412802, and D. D. acknowledges the funding by the National Program in China and startup in Fudan University.

## Author contributions

M.H. performed the experiments and analyzed data, F.W. carried out the theory and simulation, M.H., F.W., L.C., P.H., Y.L., X.G., K.L.C. and D.D. discussed the results. M.H., F.W., K.L.C. and D.D. wrote the manuscript, and D.D. supervised and guided the project.

## Competing interests

The authors declare no competing interests.
