## [Peer Review File · Nature Communications]

Near-infrared-laser-navigated dancing bubble within water via a thermally conductive interfaceREVIEWER COMMENTS

Reviewer #1 (Remarks to the Author):

In the present work, authors observe a spontaneous bubble bouncing in a frequency of tens-of-Hertz by exposing water to a high power laser. To explain the underlying physical mechanism, authors conducted a systematic investigation through high-speed imaging, numerical simulation, PIV flow velocity measurement, scaling analysis of thermal Marangoni force. The bouncing behavior as well as the followability is verified by the simulation. Moreover, some of the potential applications are also demonstrated. Overall, the work is well implemented. I would recommend the acceptance of the manuscript after considering the following comments.

1.As revealed by authors, the formation of temperature inversion layer is directly related to the high thermal conductivity of the sapphire glass. I wonder if the application of a top glass window with a low thermal conductivity will change the dynamics of bubble bouncing. Such a comparison will be very helpful.

2.Water heats up by absorbing laser energy. The location of focal point along the vertical direction somehow will influence the generation of the temperature inversion layer and hence the dynamics of bubble bouncing. Can authors put some discussions on this?

3.In the analytical expression of Gaussian laser light intensity distribution along r direction of the section "Observation of the bouncing bubble" and S3 of the supplementary materials, the symbol P should be corrected. Light intensity obviously has a unit of W/m^2 or than W. Please check and correct it.

4.In the section "Mechanism of the bubble bouncing", authors used COMSOL to obtain numerical simulation results. They reconstructed the flow field. Figure 3G clearly shows that outside the thermal BLs, they obtained the Marangoni flow with the opposite direction. However, authors also claimed that outside the thermal BL, there is not temperature gradient. Can authors explain this inconsistency?

5.The bubble size and the thickness of thermal BLs are in the same scale. As a result, when the bubble touches surface, some of it is in the BLs while the rest is in the normal temperature gradient. It's more like a competition of buoyancy force, upward thermal Marangoni force and downward Marangoni force. Did you evaluate the proportions of the individual components? Is the result consistent with what you presented in your manuscript that the upward thermal Marangoni force can be ignored.

6.In the first supplementary movie, I noticed that there is an initial lateral vibration of the bubble. Can authors explain this?

7.Among the four applications of bubbles in composition fabrication shown in Figure 5, I did not see much relevance of the particle collection one to the manuscript. I recommend authors to remove this application.

8.Hb in Figure S2 should be depicted in Figure 1. Moreover, Figure S2B was not explained everywhere. The x axis scale in Figure S2B is not the same with that in Figure S2A.

9.In the sixth line below figure 4 "the travel distance an be evaluated" "an"—> "can".

Reviewer #2 (Remarks to the Author):

The paper presents experiments, scaling law arguments and numerical simulations of a phenomenon/process where a near-infrared laser beam is used to form, grow, bounce vertically, and displace horizontally a gaz bubble in a liquid below a transparent top cover.

First, the authors present the bubble growth and bouncing phenomenology for a non-moving

laser beam and identify a range of bubble size (or of laser illumination duration) for which bouncing is observed. They provide measurements and simulations of the temperature and thermal convection flow fields induced by the laser, including in the thin temperature inversion layer below the heat-absorbent cover plate. They propose a bi-directional Marangoni-driven mechanism for the spontaneous bouncing of the bubble: downward Marangoni force due to the adverse temperature gradient below the plate and upward Marangoni force (adding up with buoyancy) due to the positive gradient in the bulk. They derive the scaling dependence of the lower and upper bounds of the bouncing bubble size range on the laser power, as well as the scaling dependence of the bouncing frequency on the bubble size, which they both compare to experimental measurements.

Second, the authors present experiments with a translating laser beam, which displaces the bubble horizontally while preserving the vertical bouncing at low displacement speeds but not at large ones. They derive an expression for the maximal translating speed at which the bubble can follow the laser, as well as a scaling expression for the transition between bouncing and non-bouncing under translation. Finally the authors show examples where the displaced bubble can trap other bubbles, droplets or solid particles that float in the liquid, by coalescing, penetrating or wetting them.

The paper presents rather exhaustive observations and analyses of a generally interesting bouncing phenomenon as well as example of a potential use of it. However, although the different observations are gradually introduced and well related to each other, many crucial points in the presentation, in some experimental information and especially in the analyses and comparisons with the experimental data, which are listed below, are lacking or deserving significant clarifications. Besides these crucial points, some of the discussions are difficult to follow, in particular technical discussions, because of syntactical issues in English (if the authors can clarify the crucial points, they should make sure to eliminate these syntactical issues). Last and importantly, the authors should clarify what is new and what is not among the many results they present.

1. Several arguments are not clear to me regarding the temperature evolution when the laser is not moving (S3).

I do not understand the scaling about the long-time temperature evolution (S3), $\Delta T \propto t^{1/2}$. For a slender cylindrical laser beam with a steady power, the radial diffusion of heat should be 2 dimensional. In the absence of advection the temperature should reach a steady value, $\sim (r_l)^2 P / \lambda$. Could the author comment and clarify which of the subsequent analyses this would modify and how?

Also, could the authors clarify why the laser-induced convection can be neglected in this analyses?

2. The definition for the time origin ($t = 0$) is not explicitly given and it actually changes between the paper and the different supplementary materials without any explicit mention (between laser starting and bubble nucleation). This seems to be confusing not only for the reader but also for the authors themselves: The scaling for the laser induced temperature field (S3) cannot be used with a Plesset-Zwick law for the bubble growth without accounting for the time shift between the bubble nucleation and the laser starting. Could the author clarify?

3. Besides the two scalings mentioned above, the authors provide many scaling analyses

but many of them are difficult or impossible to follow, because some notations are not introduced properly or not fully consistent, some expressions are not explicitly given (including in supplemental materials), and many scaling analyses only present the proportionality dependence on one parameter (e.g. laser power) while the other parameters are hidden.

a. The authors should define clearly each notation and give the actual experimental value for each of the parameters used in the analyses (for instance the value of dy/dT , and all the other ones, should be given).

b. The full dimensional expression for each scaling law should be given before a specific scaling dependence on 1 parameter is highlighted.

c. For scaling dependences on 1 single parameter, the value of the dimensional pre-factor obtained from the theoretical analyses should be systematically given and compared to the experimentally observed value. For instance in figures 3E, 3F, 4C.

4. It is not clear how the temperature measurements in fig 2A are actually obtained. Could the authors clarify how the temperature signal is quantitatively extracted from the IR camera signal and to which part of the liquid does this field actually refer to (given background thermal rays, reflexion, absorption and emissivity of the liquid)?

5. The derivation of the scaling for the thickness of the inversion layer is not clear (S7).

What is the situation really modeled for the convection velocity: a constant flux condition? applying on the surface of a cylinder? is the axial symmetry considered?

What is the dimensional factor missing in $V_b \propto Gr^{(1/2)}$?

Also the final expression for δ_{th} is found to be independent on the plate properties, which seems contradictory with the authors' observation that the thermal effusivity of the plate is important. Could the authors clarify this point?

6. On page 9, it is not clear what is the status of the 0.1 in $H_{b,cr} \sim 0.1 \cdot R$. Is this a fitting parameter or does it have a physical justification? In both cases this should be stated in the text.

7. On page 11 the derivation of eq (6) is impossible to follow. The assumptions should be clearly stated, the expression for τ_c and τ_d should be given, as well as all the necessary steps with all notations properly defined.

Also it would help the reader to summarize with simple arguments why and how the bouncing criteria differs when the laser moves relative to the case when it is static.

8. It seems obvious to the authors, though it is not explicitly justified, that the existence of a temperature inversion is sufficient to explain that the drop should bounce. However, one could also imagine that the bubble would reach a stable equilibrium position, close to the depth of the temperature maximum, where the net Marangoni force and buoyancy balance each other. It seems that the simulations presented in fig 3G and 4D do not answer to this point.

Could the authors comment?

9. Importantly, the authors should clarify what is new and what is not among the results they present.

For instance, Marangoni induced displacement of bubbles by laser illumination has already been reported (see e.g. Zhao et al, Lab Chip, 2014, 14, 384). Are equation (5) and the subsequent derivation of the maximal bubble displacement velocity original results? If not this should be explicitly stated.

The originality should be clarified at each step in the presentation or analyses the authors report.

10. Relative to the demonstration in figure 5, how much is bouncing important in coalescence with bubbles, drops or particles?

11. In the introduction, I do not understand why Lohse & Zhang 2020 (ref 11) is cited for the bouncing of drops and bubbles. This review paper does not concern this thematic.

12. In the introduction, the seminal contribution regarding drops bouncing on a vibrated bath is the work of Couder and co-workers. It should, therefore, be cited.

13. The dimension is missing in the vertical axis in figure 4C.

Reviewer #3 (Remarks to the Author):

In this paper, the authors propose the use of thermal Marangoni flows to manipulate single bubbles and droplets. The results presented are compelling, the science is interesting, the research well executed, and the paper is rather well-written. Furthermore, the supplementary information brings real added value. I have, however, concerns which I detail below that I would like the authors to address before I can recommend publication in Nature Communications.

Motivation, claims and novelty:

First, I have an issue with the motivation brought forth by the authors. In their listing of the technologies allowing for the manipulation of single bubbles or droplet, the authors mostly omit to discuss acoustical or optical tweezers which appear to be more straightforward, reliable, and impose less requirements on the sample. I do see the interest to create complex constructs as proposed in the third part of the paper.

The claim for industrial relevance (p13 and introduction) also seems difficult to support, since the technique applies to the manipulation of single bubbles/droplets.

Finally, oscillating bubbles induced by (thermal) Marangoni forces has been the object of several recent papers which is a concern for Nature Communications. Nevertheless, I do see the uniqueness of this particular physical configuration and I believe the novelty claims should be best focused on how this effect is put to use in the third part of the paper by further detailing the unique assemblies this technique can offer.

other comments/concerns:

“the observed 1/2 scaling relation of temperature evolution”. I am puzzled by this statement.

First, based on the supplementary information (Fig. S3B), the scaling in $\frac{1}{2}$ is not very convincing. In addition, $\frac{1}{2}$ for heat diffusion (disregarding flow during transient build-up) corresponds to a plane, 1D cartesian diffusion. In Fig. S3B, we can see clearly the 1 power law for short time, corresponding to a (mostly) spherical diffusion of the heat deposited in the bulk by the gaussian Beam. In this context, isn't the decrease of slope observed after 1 second simply the slow transition toward the steady state temperature profile (slope = 0) existing for spherical heat diffusion? (Note that 1 s would be consistent with the typical diffusion timescale given the size of the laser beam). In this case, the effective power law would depend on the time at which nucleation occurs.

If this is not the case, why does the power law in Fig S3B differs so significantly from $\frac{1}{2}$ and why would the authors expect $\frac{1}{2}$?

Finally, this scaling is bound to lose validity once the bubble is formed since it acts as a heat sink and changes the local absorption and light transmission. It is thus not trivial to me, based on the proposed arguments, why one would still expect a linear bubble growth over time.

P4: "The existence of temperature inversion layer is attributed to the viscous and thermal boundary layers (BLs), as swept by the thermally-driven buoyancy flow during laser heating". What is dominant: the flow or the large conductivity of the sapphire that thus evacuates much heat near its surface?

Thermal imaging: the authors do not provide much information on the thermal imaging. How do the authors solve the issue of imaging quantitatively through a significant layer of water? In particular (p15), how is the thermal imaging calibrated?

When the bubble is partly in and partly outside the boundary layer, can one simply consider 2 opposite forces, where the Maragoni force induced by the boundary layer is stronger (as suggested in the article) or does the bubble behaves as a dipole source for the flow? What would this difference entail?

P2: "By utilizing an acoustic or magnetic field, non-invasive manipulation of bubbles is achieved, but sophisticated systems or potentially harmful sample pretreatments for additional properties are required" This is not necessarily true for acoustics- or optics-based techniques.

Are the oscillations in Fig 1B real or only an effect of imperfect detection? If they are real, what is their source?

P5: "the temperature peak is confirmed to be around hundreds of micrometers below the interface". Experimentally, the inversion goes until 0.4mm, this is twice as much as in the simulation. Furthermore, the profiles differ. Where do these differences come from? Finally, the experimental curve is plot in semilog and the simulation curve has linear axes. This should be uniform. Is the temperature profile decay for large z comparable in both cases?

Fig 2E: PMMA has a lower conductivity than water. If, as the abstract suggests, thermal conductivity is key, why does it still induce a temperature inversion (albeit a small one)?

Equation 4: although the scaling obtained seems to work, I am confused by the approach: the instability and thus the oscillation frequency in both cases originate from the competition of 2 opposing forces. Without either one, there would be no frequency. In term of oscillator, it is therefore the dependence on z of the resulting force that gives rise to the existence of an eigen frequency. How correct is then this analysis? This scaling implicitly considers both forces to be equal and opposite. I think these considerations should be more clearly introduced and discussed.

P11: "the obtained maximum translation speed is consistent with the experimental observation, which is about 40 mm/s" How does this maximum translation speed depend on the laser power?

P13: "the pre-existing bubble b and c along the pathway at 0.8 s, resulting in the sequential capture and fast coalescence at 0.9 and 1.0 s. Then the combined large bubble a + b + c

continues its motion as guided by the laser (Movie S7).” I do not see the practical interest of this: there are many ways to induce coalescence, the difficulty is to prevent it.

P14: “the solutal Marangoni force is vanished, and a distinct underlying mechanism for the bouncing motion is attributed to the thermal Marangoni” This statement is confusing: from the title of the reference discussed, it appears to also consider thermal Marangoni as (partial) driving mechanism.

P16: section “PIV measurement for thermal buoyancy flow induced by laser heating”. Since the PIV measurements are only shown in SI, I do not think it should be in the methods of the article itself.

Fig 2D: the temperature profile does not seem to match Fig 2A. Why is that?

The x-axes in Fig 2 B and D should have the same limits. Furthermore, these plots appear never to reach the state where the buoyancy force overcomes the Marangoni force.

Fig 2D: why does the F_m - force keep increasing after $\Delta t_{th}/2$, while the bubble now experiences a opposite Marangoni effect?

Minor comments:

Fig 2B: is the substrate interface at $z = 0$?

Which laser power was used to calculate Fig 3 B and D?

I would be helpful if R_{up} and R_{low} are depicted in Fig 3 B and D.

Eq. 2: where do these scaling laws exactly come from.

P8, last line. The scaling for the thermal boundary layer thickness is obtained from this is a dimensional analysis with 3 nondimensional numbers. However, it entails non-trivial interconnected effects. Since the authors have thermal imaging capability, it would be interesting to verify this law experimentally.

Furthermore, why would this law remain true in the presence of a bubble whose presence changes heat deposition and induces its own mixing?

P9: “by setting $H_b;cr = 0:1R$.” What is the physical argument behind this operation?

Eq.4: why keeping the π in the dimensional analysis when the $4/3$ is already removed?

Eq. 4: Equal sign of approximately equal?

Text:

“followability”: to me, this appear not to be the best choice of word here since the authors discuss the capacity of the bubble to follow the laser rather than the capacity of the laser to be followed.

P2: ‘fascinating’. I would suggest letting the reader judge of this.

P3: “Here based on the technological advancement of laser impacting on water, by simply designing a specific thermally conductive interface, we directly realize both the vertical bouncing and the horizontal translating of the millimeter-sized bubble within pure water, achieving the dancing bubble within water in 3D.” Please rephrase.

P3: “Additionally, we demonstrate the remarkable manipulation capability of the bubble to interact with the preexisted droplet or nanoparticles” Please rephrase.

P4: “During the whole operation process around 50 seconds,” please rephrase

P4: “robust or resilient” Do the authors mean “robust and resilient”?

P4: “at $t = t_0$ ” the plots refer to $t=0$.

P6: “Such the temperature inversion might be responsible for the subsequent bouncing bubble.” Please rephrase.

P8: “two important issues are required to be addressed:” please rephrase

P9: "converted into kinetic energy of bubble with the initial velocity" please rephrase

P9: "distance an be evaluated" -> "can"

P11: "The followability of bubble with laser is driven" please rephrase

Caption Fig.5: "Steering the bubble to interplay with the bubble, droplet and particles".

Please make the plurals consistent.

P13: "This unique capability of 3D manipulation for the bubble demonstrates the potential for the versatile applications" please rephrase.

P14: "Marangoni force is vanished,"

P14: "of bubble are different in many orders of magnitude"

P17: "considering interfacial tension as a linear function with temperature"

Reviewer 1

In the present work, authors observe a spontaneous bubble bouncing in a frequency of tens-of-Hertz by exposing water to a high power laser. To explain the underlying physical mechanism, authors conducted a systematic investigation through high-speed imaging, numerical simulation, PIV flow velocity measurement, scaling analysis of thermal Marangoni force. The bouncing behavior as well as the followability is verified by the simulation. Moreover, some of the potential applications are also demonstrated. Overall, the work is well implemented. I would recommend the acceptance of the manuscript after considering the following comments.

Thanks for “the acceptance of the manuscript after addressing the comments”!

- 1. As revealed by authors, the formation of temperature inversion layer is directly related to the high thermal conductivity of the sapphire glass. I wonder if the application of a top glass window with a low thermal conductivity will change the dynamics of bubble bouncing. Such a comparison will be very helpful.*

Thanks for this wonderful comment! We have carefully elaborated the effect of the thermal conductivity of the cover materials by the following revision:

- Perform experiments for various cover materials, as shown in Supplementary Section 3 “The effect of cover materials on the bouncing behavior” with Fig. S3;
- Build model to understand temperature inversion layer, as shown in Supplementary Section 7 “Model for temperature inversion layer” with Fig. S6;
- Clarify its physical mechanism in the section of “Formation of temperature inversion layer” in the main text.

In experiments, to further verify the effect of thermal conductivity of the top glass on the formation of temperature inversion layer, we tested various glasses with different thermal conductivity (see Table S1). The bubble bouncing behavior still holds for higher thermal conductivity cover such as quartz and sapphire cover, while bouncing disappears for lower thermal conductivity cover, such as PMMA and PDMS cover. As shown in Figure S3, no bouncing is observed for the PMMA cover, and bubble just hangs at the solid-liquid interface.

Table S1: Material Properties ($T = 293.15\text{K}$)

Properties	Density	Thermal conductivity	Heat capacity	Thermal diffusivity	Thermal effusivity
Notation	ρ	k	c_p	κ	$e_i = \sqrt{k_i \rho_i c_{pi}}$
Units	kg/m^3	$\text{W}/(\text{m} \cdot \text{K})$	$\text{kJ}/(\text{kg} \cdot \text{K})$	m^2/s	$\text{W}\sqrt{\text{s}}/\text{m}^2\text{K}$
Water	1000	0.59	4.2	1.4×10^{-7}	1574.2
Sapphire	3980	23.1	0.761	7.63×10^{-6}	8364.5
Quartz	2200	1.32	0.772	7.77×10^{-7}	1497.3
PMMA	1190	0.19	1.42	1.12×10^{-7}	566.6
PDMS	970	0.16	1.46	1.13×10^{-7}	476.0

Figure S3: Snapshots and comparison for experiments with different cover materials. (A) High-speed images for the produced bubble within water for PMMA cover. (B-D) The bubble radius R , central position H_c , and topmost position H_t versus time t .

In simulation, the corresponding simulated temperature distribution shown in Fig. S6A also indicates that the thickness of temperature inversion layer becomes thinner for cover materials with a lower thermal effusivity, thus the temperature inversion layer is too thin to cause bubble bouncing for the case of PMMA and PDMS.

In theory, to qualitatively understand temperature inversion layer, we build the model to demonstrate the relevant physical parameters for the thickness of temperature inversion layer. The dependence of ratio between the thickness of temperature inversion layer (δ_{inv}) and thickness of thermal boundary layer (δ_{th}) on the ratio between thermal effusivity of cover material (e_s) and that of water (e_w) is shown in Fig. S6C. The detail derivation is provided in the revised supplementary Section 7 “Model for temperature inversion layer”.

Figure S6: Model for TIL. (A) Temperature profile for different cover materials obtained from simulation. (B) Temperature profile for sapphire cover with different thickness obtained from simulation. (C) Dimensionless thickness of TIL dependent on the thermal effusivity of cover material. (D) The scaling relation between the thickness of TIL and laser power obtained from simulation.

In the limit of high effusivity or thermal conductivity (such as sapphire), the thickness of temperature inversion layer nearly approaches that of the thermal boundary layer ($\delta_{inv} \approx \delta_{th}$). Then, the thickness of temperature inversion layer extracted from simulation agree well with the scaling relation $\delta \propto P^{-1/4}$ for the analysis of the thickness of thermal boundary layer (Fig. S6D).

2. *Water heats up by absorbing laser energy. The location of focal point along the vertical direction somehow will influence the generation of the temperature inversion layer and hence the dynamics of bubble bouncing. Can authors put some discussions on this?*

Thanks for the comment. In our experiments, the laser beam output from a 980-nm fiber coupled diode laser is first collimated by a plano-convex lens to a beam with diameter of about 2.2 mm. Then, the collimated laser beam is slightly focused by a focusing lens (L1) to a spot with diameter of about 1 mm.

The bouncing bubble is observed by either using the 2.2 mm collimated laser beam or the 1 mm slightly focused laser beam, although threshold laser intensity for bubble generation is different for the two cases.

Considering the penetration depth for 980-nm laser in water is about 2 cm, we treat the laser heating effect in our manuscript as a volumetric heating source, and assume that the location of focal point has a negligible effect on the formation of the temperature inversion layer (about 0.3 mm in our experiments).

In contrast, for a highly focused laser, the laser heating effect is treated as a point heat source, and the temperature peak is located at the focused point. The range of temperature distribution around bubble is about tens of micrometers, which is one order smaller than the thickness of temperature inversion layer in our experiments (about 0.3 mm), thus the resulting thermal Marangoni flow will trap the bubble to the focused spot [H. Takeuchi, *Heat Transfer Engineering*, 33, 234 (2012)], instead of bouncing.

In the revision, the sketch of laser in Fig. S1 is modified accordingly.

Fig. S1 Sketch for experimental setup.

3. *In the analytical expression of Gaussian laser light intensity distribution along r direction of the section “Observation of the bouncing bubble” and S3 of the supplementary materials, the symbol P should be corrected. Light intensity obviously has a unit of W/m^2 or than W . Please check and correct it.*

Thanks for the nice point! The symbol P indeed means the laser power with a unit of W in our manuscript, while laser intensity I is $P/(\pi r^2)$ with a unit of W/m^2 . In the revised supplementary (Section 5), we correct the symbol P as I , which represents the laser intensity.

4. *In the section “Mechanism of the bubble bouncing”, authors used COMSOL to obtain numerical simulation results. They reconstructed the flow field. Figure 3G clearly shows that outside the thermal BLs, they obtained the Marangoni flow with the opposite direction. However, authors also claimed that outside the thermal BL, there is not temperature gradient. Can authors explain this inconsistency?*

Thanks for the nice point! From the analysis in Fig. 3B in the main text and Section S8 in revised supplementary materials, when bubble is outside the temperature inversion layer, both the upward Marangoni force (F_m^+) and buoyancy force (F_b) work as restoring forces. In order to reveal the flow patterns related with these driving forces, we perform simulations for three cases corresponding to (A) only F_m^+ , (B) only F_b , and (C) the combined two forces.

For case (A) only F_m^+ , such as for a small bubble ($R < R_{cr}$), since the upward Marangoni force F_m^+ is much larger than buoyancy force F_b , F_b or the term of gravity in governing equation is ignored in analysis, and F_m^+ dominates the rebounding process. Correspondingly in simulation, the term of gravity in governing equation is ignored, and the positive temperature gradient outside the temperature inversion layer is assumed. The simulated flow pattern is obtained as shown in the following Figure (A).

For case (B) only F_b , such as for a large bubble ($R > R_{cr}$), since buoyancy force F_b is much larger than the upward Marangoni force F_m^+ , F_m^+ is ignored in analysis and F_b dominates the rebounding process. In simulation, the term of gravity in governing equation is included, while the temperature gradient outside the temperature inversion layer is neglected. The simulated flow pattern is obtained as shown in the following Figure (B), which is the Fig. 3G in the previous main text.

From the comparison between Figure (A) and (B), the flow pattern around bubble is similar, though the mechanism is totally different. For Figure (A), the convective flow is induced by the thermal Marangoni flow at the bubble interface, while for Figure (B), the flow pattern is induced by the buoyancy force driven flow associated with the bubble motion.

For case (C) the combined two forces, such as for a medium bubble ($R \approx R_{cr}$), both buoyancy force F_b and the upward Marangoni force F_m^+ should be considered. In simulation, we not only consider the term of gravity in governing equation, but also

introduce a positive temperature gradient outside the temperature inversion layer. The simulated flow pattern is shown in the following figure (C), which is Fig. 3G in the revised main text.

The details of simulation settings and description for bubble bouncing are provided in the Section S11 in the revised supplementary materials.

Figure: Simulated flow pattern for three cases dependent on the dominated forces in the rebounding process. (A) only F_m^+ , (B) only F_b , and (C) the combined F_m^+ and F_b forces.

5. *The bubble size and the thickness of thermal BLs are in the same scale. As a result, when the bubble touches surface, some of it is in the BLs while the rest is in the normal temperature gradient. It's more like a competition of buoyancy force, upward thermal Marangoni force and downward Marangoni force. Did you evaluate the proportions of the individual components? Is the result consistent with what you presented in your manuscript that the upward thermal Marangoni force can be ignored.*

Thanks for the nice comment! When some part of the bubble in the temperature inversion layer and the other part in the normal temperature gradient, there exists a competition of buoyancy force F_b , upward Marangoni force F_m^+ and downward Marangoni force F_m^- . Hence, in the revised Fig. 3D in the revised manuscript, we carefully compare the magnitude of these three forces: buoyancy force, upward thermal Marangoni force and downward Marangoni force.

For the case that bubble touches surface (as shown in Fig. 3C), from the view of scaling, the magnitude of the downward Marangoni force $F_m^- = \Delta\gamma \cdot R = \frac{d\gamma}{dT} \frac{dT^-}{dz} \cdot \min(2R, \delta_{inv}) \cdot \pi R$, where dT^-/dz for the temperature gradient within temperature

inversion layer is about 30 K/mm. The magnitude of the upward Marangoni force $F_m^+ = \Delta\gamma \cdot R = \frac{d\gamma}{dT} \frac{dT^+}{dz} \cdot \max(0, 2R - \delta_{inv}) \cdot \pi R$, where dT^+/dz for the temperature gradient outside temperature inversion layer is about 1 K/mm. Thus, when the bubble touches surface, the downward Marangoni force is always at least one order larger than the upward Marangoni force. Therefore, to simplify the analysis, the upward Marangoni force can be reasonably neglected.

In the revised manuscript, we have presented the magnitude of the upward Marangoni force (F_m^+) in Fig. 3D to compare the individual component clearly.

Fig. 3D Sketch of the competing buoyancy force and the thermal Marangoni force, and their magnitudes dependent on bubble radius

6. *In the first supplementary movie, I noticed that there is an initial lateral vibration of the bubble. Can authors explain this?*

Thanks for the comment! Indeed, we also notice the lateral vibration at the early stage when the bubble radius is small.

A possible explanation for the lateral vibration is due to the Gaussian distribution of laser intensity, where the temperature peak along horizontal direction is located at center of the laser spot. The mass center of bubble tends to match the location of temperature peak, which is also the mechanism of bubble moving with the translating laser spot. However, for a stationary laser spot, bubble may be disturbed by the buoyancy flow of water, or the fluctuation of bubble radius, or the thermal-insulate effect of the bubble, to deviate from the center of the laser spot. Then, it appears to be lateral vibration.

In future, we will further study the lateral vibration with higher spatio-temporal resolution to characterize the exact motion of bubble at the early stage, and explore its underlying physical mechanism. In this work, we mainly focus on the periodic bouncing motion of bubble along the vertical direction.

7. *Among the four applications of bubbles in composition fabrication shown in Figure 5, I did not see much relevance of the particle collection one to the manuscript. I recommend authors to remove this application.*

Thanks for the comment! In literatures, bubble motion generally is passively manipulated along the translation direction of a laser beam, but in this work, we report the periodic bouncing motion of bubble along the penetration direction of laser beam. Besides the bouncing motion along the vertical direction, the floating bubble presented in this work can translate horizontally very fast with the laser spot (about 4 cm/s), which will enhance the efficiency and accuracy of particle collection.

Due to the extended moving dimension of bouncing bubble and the Marangoni convection around the bubble, the dancing bubble has a unique advantage in particle collection and transportation, hence in a fashion similar to “deep surface cleaning”, particles can be collected or trapped by the bubble within a much larger volume (related with the vertical bouncing and horizontal translation). This novel capability of particle collection and transportation is demonstrated in the revised Supplementary Movie S10. Clearly, this remarkable particle enrichment and 3D manipulation of the bubble has implications for applications such as wastewater treatment or targeted drug delivery. Compared with other methods to trap particles, such as acoustically activated bubbles or previous optical manipulated bubbles, the underwater spherical dancing bubble in our system shows an obvious superiority in terms of enrichment concentration and flexible trajectory.

8. *H_b in Figure S2 should be depicted in Figure 1. Moreover, Figure S2B was not explained everywhere. The x axis scale in Figure S2B is not the same with that in Figure S2A.*

Thanks for the comment. In the revised manuscript, H_b is labeled as H_t , which is depicted in Fig. 1A. The description of Figure S2B is added in the revised supplementary (S2), also the x axis scale in Figure S2B is revised to be consistent with Figure S2A.

The following sentence is added in Supplementary Section S2 “Reproducibility of bouncing behavior” --- “The topmost position of the bubble H_t (as depicted in Fig. 1A, the vertical distance from the topmost of the bubble to the solid cover) is presented in Fig. S2B. The periodical displacement of the bubble implies the occurrence of the bouncing behavior, including the bouncing onset and stop moment, and the bouncing amplitude.”

9. In the sixth line below figure 4 “the travel distance an be evaluated” “an”-> “can”.

Thanks! This typo has been corrected, “the travel distance can be evaluated”.

Reviewer 2

The paper presents experiments, scaling law arguments and numerical simulations of a phenomenon/process where a near-infrared laser beam is used to form, grow, bounce vertically, and displace horizontally a gas bubble in a liquid below a transparent top cover.

First, the authors present the bubble growth and bouncing phenomenology for a non-moving laser beam and identify a range of bubble size (or of laser illumination duration) for which bouncing is observed. They provide measurements and simulations of the temperature and thermal convection flow fields induced by the laser, including in the thin temperature inversion layer below the heat-absorbent cover plate. They propose a bi-directional Marangoni-driven mechanism for the spontaneous bouncing of the bubble: downward Marangoni force due to the adverse temperature gradient below the plate and upward Marangoni force (adding up with buoyancy) due to the positive gradient in the bulk. They derive the scaling dependence of the lower and upper bounds of the bouncing bubble size range on the laser power, as well as the scaling dependence of the bouncing frequency on the bubble size, which they both compare to experimental measurements.

Second, the authors present experiments with a translating laser beam, which displaces the bubble horizontally while preserving the vertical bouncing at low displacement speeds but not at large ones. They derive an expression for the maximal translating speed at which the bubble can follow the laser, as well as a scaling expression for the transition between bouncing and non-bouncing under translation.

Finally the authors show examples where the displaced bubble can trap other bubbles, droplets or solid particles that float in the liquid, by coalescing, penetrating or wetting them.

The paper presents rather exhaustive observations and analyses of a generally interesting bouncing phenomenon as well as example of a potential use of it.

Thanks for nice summary! We appreciate that “the paper presents rather exhaustive observations and analyses of a generally interesting bouncing phenomenon”!

However, although the different observations are gradually introduced and well related to each other, many crucial points in the presentation, in some experimental information and especially in the analyses and comparisons with the experimental data, which are listed below, are lacking or deserving significant clarifications.

Besides these crucial points, some of the discussions are difficult to follow, in particular technical discussions, because of syntactical issues in English (if the authors can clarify the crucial points, they should make sure to eliminate these syntactical issues).

Last and importantly, the authors should clarify what is new and what is not among the many results they present.

Thanks for these constructive comments! These points will be addressed accordingly as below.

- Several arguments are not clear to me regarding the temperature evolution when the laser is not moving (S3). I do not understand the scaling about the long-time temperature evolution (S3), $\Delta T \propto t^{1/2}$. For a slender cylindrical laser beam with a steady power, the radial diffusion of heat should be 2 dimensional. In the absence of advection the temperature should reach a steady value, $\sim (r_l)^2 P/\lambda$. Could the author comment and clarify which of the subsequent analyses this would modify and how? Also, could the authors clarify why the laser-induced convection can be neglected in this analyses?

Thanks for the comment. For a slender cylindrical laser beam with a steady power irradiating water, the temperature evolution has two limits: $\Delta T \propto Pt$ for short time period by ignoring thermal diffusion (Fig. S4B and Fig. S4C), while $\Delta T \propto Pr_l^2/k$ for long time period by considering the balance of laser heating flux and thermal conductive flux.

Fig. S4 Measured temperature during laser impacting on water. (A) Evolution of peak temperature (T_{peak}) and the corresponding location δ_{inv} ($P=15$ W with bubble formation) from the thermal images. (B, C) During the initial pre-heating stage, the evolution of elevated temperature (ΔT) for $P=15$ W with and $P=10$ W without bubble formation.

Actually, in previous supplementary materials, the long-time temperature evolution is used to explain the almost linear growth of bubble radius, but it does not affect other analysis for bubble motion. In the revision, in order to better focus on the bouncing behavior in this work, the analysis of 1/2 scaling of long-time temperature evolution and the linear growth of bubble are removed from the main text and supplementary materials in the revision, and the explanation of bubble motion is not affected.

By comparing the trivial solution for thermal conduction $\Delta T = \frac{A_0 P}{\rho c_p} e^{-r^2/r_i^2} e^{-\alpha z} \tau$ (Eq. S4) with the simulation considering convection, as shown in Fig. R1, the temperature distribution outside the temperature inversion layer is well described by the trivial solution, while within the temperature inversion layer, the cooling effect of solid cover is dominant for our experiments. Hence, for the temperature evolution here, the laser-induced convection in the analysis for short time period can be ignored for simplicity.

Fig. R1 Comparison of the temperature distribution along Z direction at $r = 0$

2. *The definition for the time origin ($t = 0$) is not explicitly given and it actually changes between the paper and the different supplementary materials without any explicit mention (between laser starting and bubble nucleation). This seems to be confusing not only for the reader but also for the authors themselves: The scaling for the laser induced temperature field (S3) cannot be used with a Plesset-Zwick law for the bubble growth without accounting for the time shift between the bubble nucleation and the laser starting. Could the author clarify?*

Thanks for the nice comments! To avoid this unnecessary confusion, we present a consistent and clear definition of time origin in the revised supplementary materials. We define the time origin ($t = 0$) for the moment right before the bubble is visible, and the time for turning on laser during laser heating ($t_{laser\ on}$), and we refine the related figures and descriptions in the revised main text and supplementary materials.

The different definition of time origin (the time shift between bubble nucleation and laser starting) can affect the scaling relation of temperature evolution. Besides the time shift effect, once the bubble forms, the thermal-insulation effect of bubble and the mixing effect of bubble motion also affect the temperature evolution as shown in Fig. S4A after the time $t = 0$ s. Thus, we decide to remove the analysis of $1/2$ scaling of long-time temperature evolution and the linear growth of bubble in the main text and supplementary materials. And the removed parts do not affect the explanation of bubble motion, which is the core part of this work. We will further study the temperature evolution and bubble growth characteristic in future.

3. *Besides the two scaling mentioned above, the authors provide many scaling analyses but many of them are difficult or impossible to follow, because some notations are not introduced properly or not fully consistent, some expressions are not explicitly given (including in supplemental materials), and many scaling analyses only present the proportionality dependence on one parameter (e.g. laser power) while the other parameters are hidden.*
 - a. *The authors should define clearly each notation and give the actual experimental value for each of the parameters used in the analyses (for instance the value of dy/dT , and all the other ones, should be given).*
 - b. *The full dimensional expression for each scaling law should be given before a specific scaling dependence on one parameter is highlighted.*
 - c. *For scaling dependence on one single parameter, the value of the dimensional prefactor obtained from the theoretical analyses should be systematically given and compared to the experimentally observed value. For instance in figures 3E, 3F, 4C.*

Thanks for the nice comment. In the revision, we have presented a clear definition for each notation, described each expression or parameters in detail, and added the value of parameters and their explicit and detailed expressions.

- a. Definition and value for notations used in analysis is provided in Table S1 and Table S2 the Supplementary.

Table S1: Material Properties ($T = 293.15\text{K}$)

Properties	Density	Thermal conductivity	Heat capacity	Thermal diffusivity	Thermal effusivity
Notation	ρ	k	c_p	κ	$e_i = \sqrt{k_i \rho_i c_{pi}}$
Units	kg/m^3	$\text{W}/(\text{m} \cdot \text{K})$	$\text{kJ}/(\text{kg} \cdot \text{K})$	m^2/s	$\text{W}\sqrt{\text{s}}/\text{m}^2\text{K}$
Water	1000	0.59	4.2	1.4×10^{-7}	1574.2
Sapphire	3980	23.1	0.761	7.63×10^{-6}	8364.5
Quartz	2200	1.32	0.772	7.77×10^{-7}	1497.3
PMMA	1190	0.19	1.42	1.12×10^{-7}	566.6
PDMS	970	0.16	1.46	1.13×10^{-7}	476.0

Table S2: Definition of notation

Notation	Definition	Unit	Value
$d\gamma/dT$	Gradient of surface tension of water	$\text{kg}/(\text{s}^2 \cdot \text{K})$	-2×10^{-4}
β	Thermal expansion coefficient of water	1/K	2×10^{-4}
α	Attenuation coefficient of laser in water	1/m	45
ρ	Density of water	kg/m^3	998 (293 K), 958 (373 K)
c_p	Heat capacity of water	$\text{kJ}/(\text{kg} \cdot \text{K})$	4.18 (293 K), 4.22 (373 K)
k	Thermal conductivity of water	$\text{W}/(\text{m} \cdot \text{K})$	0.60 (293 K), 0.68 (373 K)
μ	Dynamic viscosity of water	$\text{mPa} \cdot \text{s}$	1.00 (293 K), 0.28 (373 K)
Pr	Prandtl number of water	1	6.99 (293 K), 1.76 (373 K)

- b. The full dimensional derivation for each scaling laws are provided in the revised Supplementary Materials, including Section 6-9. Specifically, Section 6 is for “Model for velocity and thermal boundary layer”; Section 7 for “Model for temperature inversion layer”; Section 8 for “Model for bubble bouncing”; and Section 9 for “Model for the dancing bubble”.
- c. Comparison between theoretical estimation and experimental value for dimensional prefactors and physical parameters.

Notation	Definition	Theory	Experiment	Section
V_b	Velocity of buoyancy flow	8.9 mm/s	10 mm/s	Section S6
δ_{inv}	Thickness of temperature inversion layer	0.33 mm	0.3 mm	Section S7
R_{up}	Upper bound radius for bubble bounce ($P=15$ W)	0.69 mm	0.75 ± 0.07 mm	Section S8
R_{low}	Lower bound radius for bubble bounce ($P=15$ W)	0.45 mm	0.37 ± 0.07 mm	Section S8
C_1	Prefactor in expression for frequency dominant by F_m^+	$10 \text{ mm}^{1/2}/\text{s}$	$16 \text{ mm}^{1/2}/\text{s}$	Section S8
f	Frequency dominant by F_b ($R=0.9$ mm)	19.6 Hz	19.2 ± 2.17 Hz	Section S8
$P^{3/4}/R_{up}^2$	Criterion for bubble dancing	$16.0 \text{ W}^{3/4}/\text{mm}^2$	$10.5 \text{ W}^{3/4}/\text{mm}^2$	Section S9

4. *It is not clear how the temperature measurements in fig 2A are actually obtained. Could the authors clarify how the temperature signal is quantitatively extracted from the IR camera signal and to which part of the liquid does this field actually refer to (given background thermal rays, reflexion, absorption and emissivity of the liquid)?*

Thanks for the nice comment! More details of the temperature measurements are provided here. In order to gather accurate temperature measurements, the temperature detection of the superheated liquid from the side view is calibrated, since the side wall of the quartz cuvette affects the infrared intensity signal to be detected by the camera from the target liquid surface.

Firstly, the radiometric thermal camera is so close to the target surface (less than 20 cm) that the atmospheric transmission factors can be neglected. Secondly, since the surface emissivity of both the quartz glass ($\epsilon=0.93$) and liquid water ($\epsilon=0.96$) is large (greater than 0.90), the impact of background temperature reflection can also be ignored. Thirdly, we calibrate the measured temperature from the IR camera by using a correction factor. The factor is obtained by simultaneously using the thermal camera and a thermocouple to measure the temperature evolution of a cooling process of hot glycerol ($\epsilon=0.96$, the same emissivity as that of water) in the cuvette within the temperature range from 140 °C to 30 °C. And the location of the thermocouple is exactly the same with the laser focus spot, which is very near to the quartz wall (about several millimeter). Hence, the calibrated temperature refers to the part of the irradiated liquid in the plane of laser transmission.

A new section of “Temperature measurement” is added in Materials and Methods in the main text. “An infrared thermal camera (FLIR A6750sc) was used to acquire the temperature evolution of liquid during laser irradiating. For the top view temperature detection, the sapphire glass with high transmissivity @185-5000 nm was adopted as the cover glass since infrared light can optically penetrate the sapphire glass. Since the side wall of the quartz cuvette affects the detection of infrared intensity signal emitted from the target liquid surface, the measured temperature values from the side view by the thermal camera was required to be calibrated. For the calibration, a thermocouple was exactly located at the same position of laser focus spot, closer to the quartz wall (about several millimeter). By using the thermal camera and the thermocouple to simultaneously measure the temperature evolution of a cooling process for a glycerol in the cuvette within the temperature range from 140 °C to 30 °C (the same emissivity with that of water, $\epsilon = 0.96$), the elevated temperature of the irradiated liquid in the plane of laser transmission was quantitatively calibrated.”

5. *The derivation of the scaling for the thickness of the inversion layer is not clear (S7). What is the situation really modeled for the convection velocity: a constant flux condition? applying on the surface of a cylinder? is the axial symmetry considered? What is the dimensional factor missing in $Vb \propto Gr^{(1/2)}$? Also the final expression for δ_{th} is found to be independent on the plate properties, which*

seems contradictory with the authors' observation that the thermal diffusivity of the plate is important. Could the authors clarify this point?

Thanks for this nice comment! We will clarify these points, including the effect of thermal conductivity of cover materials.

- I. For the model for the velocity and thermal boundary layer, detail derivation is provided in the revised supplementary Section S6 “Model for velocity and thermal boundary layer”.

To model the boundary layer induced by the buoyancy flow sweeping the cover surface, we first consider the buoyancy flow similar to the analysis for laminar natural convection on a heated vertical surface [*Fundamentals of Heat and Mass Transfer (John Wiley & Sons, 2011)*], which is an axisymmetric model. The expression of buoyancy flow velocity is obtained $V_b = \sqrt{g\beta\Delta TL} = \frac{\mu}{\rho L} Gr^{1/2}$. Then, we use the hydrodynamic solution by Blasius for laminar flow over isothermal plate [*Fundamentals of Heat and Mass Transfer (John Wiley & Sons, 2011)*] to obtain the expression of velocity and thermal boundary layer thickness.

- II. We have carefully elaborated the effect of the thermal conductivity of the cover materials by the following revisions:

- Perform experiments for various cover materials, as shown in Supplementary Section 3 “The effect of cover materials on the bouncing behavior” with Fig. S3;
- Build model to understand temperature inversion layer, as shown in Supplementary Section 7 “Model for temperature inversion layer” with Fig. S6;
- Clarify its physical mechanism in the section of “Formation of temperature inversion layer” in the main text.

In experiments, to further verify the effect of thermal conductivity of the top glass on the formation of temperature inversion layer, we tested various glasses with different thermal conductivity (see Table S1). The bubble bouncing behavior still holds for higher thermal conductivity cover such as quartz and sapphire cover, while bouncing disappears for lower thermal conductivity cover, such as PMMA and PDMS cover. As shown in Figure S3, no bouncing is observed for the PMMA cover, and bubble just hangs at the solid-liquid interface.

Table S1: Material Properties ($T = 293.15\text{K}$)

Properties	Density	Thermal conductivity	Heat capacity	Thermal diffusivity	Thermal effusivity
Notation	ρ	k	c_p	κ	$e_i = \sqrt{k_i \rho_i c_{pi}}$
Units	kg/m^3	$\text{W}/(\text{m} \cdot \text{K})$	$\text{kJ}/(\text{kg} \cdot \text{K})$	m^2/s	$\text{W}\sqrt{\text{s}}/\text{m}^2\text{K}$
Water	1000	0.59	4.2	1.4×10^{-7}	1574.2
Sapphire	3980	23.1	0.761	7.63×10^{-6}	8364.5
Quartz	2200	1.32	0.772	7.77×10^{-7}	1497.3
PMMA	1190	0.19	1.42	1.12×10^{-7}	566.6
PDMS	970	0.16	1.46	1.13×10^{-7}	476.0

Figure S3: Snapshots and comparison for experiments with different cover materials. (A) High-speed images for the produced bubble within water for PMMA cover. (B-D) The bubble radius R , central position H_c , and topmost position H_t versus time t .

In simulation, the corresponding simulated temperature distribution shown in Fig. S6A also indicates that the thickness of temperature inversion layer becomes thinner for cover materials with a lower thermal effusivity, thus the temperature inversion layer is too thin to cause bubble bouncing for the case of PMMA and PDMS.

In theory, to qualitatively understand temperature inversion layer, we build the model to demonstrate the relevant physical parameters for the thickness of temperature inversion layer. The dependence of ratio between the thickness of temperature inversion layer (δ_{inv}) and thickness of thermal boundary layer (δ_{th}) on the ratio between thermal effusivity of cover material (e_s) and that of water (e_w) is shown in Fig. S6C. The detail derivation is provided in the revised supplementary Section 7 “Model for temperature inversion layer”.

In the limit of high effusively or thermal conductivity (such as sapphire), the thickness of temperature inversion layer nearly approaches that of the thermal boundary layer ($\delta_{inv} \approx \delta_{th}$). Then, the thickness of temperature inversion layer extracted from simulation agree well with the scaling relation $\delta \propto P^{-1/4}$ for the analysis of the thickness of thermal boundary layer (Fig. S6D).

Figure S6: Model for TIL. (A) Temperature profile for different cover materials obtained from simulation. (B) Temperature profile for sapphire cover with different thickness obtained from simulation. (C) Dimensionless thickness of TIL dependent on the thermal effusivity of cover material. (D) The scaling relation between the thickness of TIL and laser power obtained from simulation

6. On page 9, it is not clear what is the status of the 0.1 in H_b , $cr \sim 0.1 \cdot R$. Is this a fitting parameter or does it have a physical justification? In both cases this should be stated in the text.

Thanks for the comment. As shown in Fig. 1C, we note that the bouncing begins with a small amplitude, to identify the lower threshold of bubble bouncing, a criterion for the bubble vertical displacement H_t is needed. Thus, to eliminate the effect of fluctuation of bubble radius and lateral oscillation of bubble, we set the low bouncing threshold of $H_{t,cr} \approx 0.1R_{low}$ that only when the bubble vertical displacement is comparable with 0.1 of its radius, based on the argument that bouncing becomes observable by high-speed images feasibly. The detail derivation is provided in the revised Supplementary Section 8.

7. On page 11 the derivation of eq (6) is impossible to follow. The assumptions should be clearly stated, the expression for τ_c and τ_d should be given, as well as all the necessary steps with all notations properly defined. Also it would help the reader to summarize with simple arguments why and how the bouncing criteria differs when the laser moves relative to the case when it is static.

Thanks for this comment! We have provided more details for equation (2) and equation (6) in the revised manuscript.

For equation (2) on page 8 in section of “Mechanism of the bubble bouncing”, the related expressions are provided as below.

“When the bubble is outside the temperature inversion layer (Fig. 3A), both the buoyancy force $F_b = \frac{4}{3}\pi\rho gR^3 \propto R^3$, and the upward Marangoni force $F_m^+ = \Delta\gamma \cdot R = \gamma_{th}^+ \cdot 2\pi R^2 \propto R^2$ ($\gamma_{th}^\pm = \frac{d\gamma}{dT} \frac{dT^\pm}{dz}$ is the gradient of surface tension) serve as the restoring force, bringing bubble to the solid wall as expected. Once the bubble is mostly immersed in the temperature inversion layer (Fig. 3C), the buoyancy force F_b is the same, and the upward Marangoni force $F_m^+ = \gamma_{th}^+ \cdot \max(0, 2R - \delta_{inv}) \cdot \pi R$ depends on the height outside the temperature inversion layer. However, the downward Marangoni force $F_m^- = \gamma_{th}^- \cdot \min(2R, \delta_{inv}) \cdot \pi R$ becomes significant, enabling the possibility to push bubble downward away from the solid wall. From this scaling analysis, the comparison of the magnitude for these forces is shown in Fig. 3B and Fig. 3D.

Regarding this bouncing behavior, the upper bound radius (R_{up}) can be evaluated by balancing buoyancy force (F_b) with the downward Marangoni force (F_m^-),

$$R_{up}^2 = \frac{3\gamma_{th}^-}{4\rho g} \delta_{inv}, \quad (\text{Eq. 2})$$

where the gradient of surface tension $\gamma_{th}^- \propto \Delta T \propto P$ (Supplementary Section 5), and the thickness of temperature inversion layer $\delta_{inv} \propto P^{-1/4}$ (Supplementary Section 7), then $R_{up} \propto P^{3/8}$.”

For equation (6) on page 11 in section of “Bubble dancing with horizontal translation”, the related expressions are provided as below.

“Besides the dwelling timescale $\tau_d = 2r_l/v_l$, τ_c is the characteristic timescale for the heat transfer (defined in Fig. S4). For the short time period ($\tau_d < \tau_c$), the elevated temperature of water (ΔT) increases with time ($\Delta T \propto P\tau_d$, Supplementary Section 5), while for the long time period ($\tau_d > \tau_c$), the elevated temperature of water reaches a steady state ($\Delta T \propto P\tau_c$).

By substituting the scaling relation of $\gamma_{th}^- \propto \Delta T$ into the previous analysis (Eq. 2), the upper bound radius for dancing bubble can be obtained, $R_{up}^2 = 3\gamma_{th}^- \delta_{inv} / 4\rho g \propto P^{3/4} \min(\tau_d, \tau_c)$.

Then, the following scaling is obtained,

$$P^{3/4}/R_{up}^2 \propto [\min(\tau_d, \tau_c)]^{-1} \sim \begin{cases} \tau_c^{-1} \sim \text{const.}, & (\tau_d > \tau_c, v_l < 2r_l/\tau_c) \\ \tau_d^{-1} \sim v_l, & (\tau_d < \tau_c, v_l > 2r_l/\tau_c) \end{cases}$$

which agrees excellently with experiments (Fig. 4C).”

8. *It seems obvious to the authors, though it is not explicitly justified, that the existence of a temperature inversion is sufficient to explain that the drop should bounce. However, one could also imagine that the bubble would reach a stable equilibrium position, close to the depth of the temperature maximum, where the net Marangoni force and buoyancy balance each other. It seems that the simulations presented in fig 3G and 4D do not answer to this point. Could the authors comment?*

Thanks for this insightful comment! By fixing the temperature inversion layer in theory, bubble will finally tend to reach a stable equilibrium position. However, in our experiments, the formation of temperature inversion layer is associated with laser heating effect and convection flow, and might be disturbed by the thermal insulation effect of bubble and mixing effect induced by bubble bouncing. So, the temperature inversion layer actually experiences a construction-destruction-reconstruction process during bubble motion.

Although the laser is continuous, the downward Marangoni force impacted on bubble is in a fashion like a short impulse [similar idea to Zeng, et al, Proc. Natl. Acad. Sci. U. S. A. 118 (2021)]. In the simulation as shown in Fig 3G and 4D, the temperature inversion layer is introduced by a step function with time and only lasts for a short pulse ($0 < t < 5$ ms). By considering the stable temperature inversion layer and the short impulse acted temperature inversion layer, the simulated bubble motions are compared in the following figure. If the temperature inversion layer exists continuously or is fixed, the bubble reaches a stable equilibrium position in the bulk water.

Simulation details are provided in Sect 11 in the revised supplementary materials.

Figure: The simulated bubble motion for two cases. (A) The temperature inversion layer acts as a short impulse on bubble motion. (B) The temperature inversion layer is fixed during the whole bubble motion process.

9. *Importantly, the authors should clarify what is new and what is not among the results they present. For instance, Marangoni induced displacement of bubbles by laser illumination has already been reported (see e.g. Zhao et al, Lab Chip, 2014, 14, 384). Are equation (5) and the subsequent derivation of the maximal bubble displacement velocity original results? If not this should be explicitly stated. The originality should be clarified at each step in the presentation or analyses the authors report.*

Thanks for the comment!

The main originality or novelty of this work is briefly summarized as following:

- (1) We observe the vertical bouncing bubble in water during laser impacting on water.
- (2) We propose the underlying physical mechanism based on the unique temperature inversion layer, which is related with the high thermal conductivity of the cover glass.
- (3) We show the bouncing bubble can be steered horizontally by the laser, resulting in the dancing bubble.
- (4) We perform the scaling law analysis and numerical simulation, and the theory has a remarkable agreement with experiments.

In terms of Marangoni induced displacement of bubbles, previous work on bubble manipulation by laser is focused on bubble following movement with the translating laser spot or bubble trapping by a focused laser spot. Here, by designing the temperature field along laser propagation direction, we have successfully realized the bubble bouncing behavior, and this vertical bouncing behavior in water extends a new dimension along the vertical direction for bubble manipulation by laser.

In this work, we find bubble can bounce in water periodically along the direction of laser propagation, by only building a non-monotonic temperature distribution via laser heating. Moreover, by utilizing the horizontal motion of laser spot, we clearly demonstrate the 3D sophisticated manipulation of bubble or “dancing bubble” in water for the first time to the best of our knowledge.

As for the theoretical analysis, the full expression derivation and details are listed in the revised Supplementary Section 5-9, and the results referenced from previous work have been marked citations.

10. *Relative to the demonstration in figure 5, how much is bouncing important in coalescence with bubbles, drops or particles?*

Thanks for the comment! This work presents the bouncing bubble in water arising from the temperature inversion layer. This dancing bubble has two remarkable features, one is bouncing, and the other is steerability.

In the revised Fig 5, Fig5A and Fig 5B demonstrate these two features simultaneously, allowing 3D motion and leaping over a wall. Fig 5C-E mainly show its steerability to interact with the preexisting objects including walls, bubbles, droplets or nanoparticles, although bouncing might not be necessarily related.

Nevertheless, due to the extended moving dimension of bouncing bubble and the Marangoni convection around the bubble, the dancing bubble has a unique advantage in particle collection and transportation (Fig 5E) taken as an example, hence in a fashion similar to “deep surface cleaning”, particles can be collected or trapped by the bubble within a much larger volume (related with the vertical bouncing and horizontal translation). This novel capability of particle collection and transportation is demonstrated in the revised Supplementary Movie S10. Clearly, this remarkable particle enrichment and 3D manipulation of the bubble has implications for applications such as wastewater treatment or targeted drug delivery. Compared with other methods to trap particles, such as acoustically activated bubbles or previous optical manipulated bubbles, the underwater spherical dancing bubble in our system shows an obvious superiority in terms of enrichment concentration and flexible trajectory.

11. In the introduction, I do not understand why Lohse & Zhang 2020 (ref 11) is cited for the bouncing of drops and bubbles. This review paper does not concern this thematic.

Thanks for suggestions! This reference has been updated as below, “Particularly, arising from the effect of physicochemical hydrodynamics in a binary liquid (ethanol and water) (18)”.

12. In the introduction, the seminal contribution regarding drops bouncing on a vibrated bath is the work of Couder and co-workers. It should, therefore, be cited.

Thanks for suggestion! The following two references have been cited accordingly-- “such as droplets bouncing on a vibrating fluid bath (13–15)”.

[14] Y. Couder, E. Fort, C.-H. Gautier, A. Boudaoud, From Bouncing to Floating: Noncoalescence of Drops on a Fluid Bath. *Physical Review Letters*, 94, 177801 (2005).

[15] Couder, S. Protière, E. Fort, A. Boudaoud, Walking and orbiting droplets. *Nature*, 437, 208 (2005).

Review 3

In this paper, the authors propose the use of thermal Marangoni flows to manipulate single bubbles and droplets. The results presented are compelling, the science is interesting, the research well executed, and the paper is rather well-written. Furthermore, the supplementary information brings real added value.

Thanks for acknowledging “The results presented are compelling, the science is interesting, the research well executed, and the paper is rather well-written.”

I have, however, concerns which I detail below that I would like the authors to address before I can recommend publication in Nature Communications.

Thanks for the insightful questions, which will be addressed as below.

Motivation, claims and novelty:

First, I have an issue with the motivation brought forth by the authors. In their listing of the technologies allowing for the manipulation of single bubbles or droplet, the authors mostly omit to discuss acoustical or optical tweezers which appear to be more straightforward, reliable, and impose less requirements on the sample. I do see the interest to create complex constructs as proposed in the third part of the paper.

Thanks for this comment! This work mainly presents a physical phenomenon of bouncing bubble and dancing bubble during laser impacting on water underneath a glass cover, and proposes the underlying physical mechanism through scaling analysis and numerical simulation.

The comparisons with other technologies, acoustical and optical tweezers, will be discussed here. Firstly, optical tweezers have long been used as a powerful tool to trap high-index particles [A. Ashkin, Opt. Lett. 11, 288 (1986)], which rely on the extremely high gradient force produced near the beam waist of a tightly focused laser beam. However, a particle with a refractive index lower than its surrounding medium (a bubble being the limiting case) should be repulsed from the focus of a Gaussian beam. Although attempts have been made to overcome this limitation of conventional optical tweezers for trapping low-index particles by designing an optical vortex beam [K. T. Gahagan, Opt. Lett. 21, 827 (1996)] or using a two-dimensional interference pattern [M. P. MacDonald, Opt. Lett. 26, 863 (2001)], manipulation of bubbles in liquid by optical tweezers still remains challenging.

Secondly, acoustic methods are regarded as a promising alternative to trap and manipulate objects. Especially, the surface acoustic wave (SAW)-based microfluidic device has proven to be a potential powerful platform for manipulating bubbles, droplets and particles. In one-dimensional standing waves formed by two opposite travelling SAWs, the pressure gradient traps the bubbles at the potential wells, either the pressure nodes or pressure antinodes, according to the resonance frequency of

bubbles with the respect to the acoustic frequency. Bubble position can be controlled by dynamically changing the position of the potential wells. Through designing a two-dimensional standing surface acoustic waves with two pairs of transducers, the trajectories of bubbles can be configured in square or circular tracks [Long Meng, Appl. Phys. Lett. 100, 173701 (2012)]. However, technological limitations still exist for precise control of bubble location in three dimensions via these acoustic-based methods [Ding, Lab Chip, 2013, 13, 3626]. The manipulation in the out-of-plane (z) direction needs the innovation of device design and the improvement of the performance.

Additionally, a hybrid bubble-based manipulation technique called as “optoacoustic tweezers” for precise and on-demand handling of micro-objects by synergistically combining optothermal and acoustical strategies has been proposed [Xie, Lab Chip, 2013, 13, 1772]. This technique utilizes the optothermal effect to generate size and location controllable bubbles and acoustic radiation forces to trap particles/cells.

Here, in our work, by only using the optothermal effect, the generation and 3D manipulation of bubbles, as well as particles trapping or concentrating are realized. Unlike optical or optoacoustic tweezers, our experimental setup does not require high-numerical aperture lenses or sophisticated systems. All setups are easy-to assemble and easy-to operate.

The comparison between these technologies has been added in the discussion part on page 14 in the revised main text, as below,

“Conventionally, by utilizing the applied external stimuli (such as magnetic, electric, acoustic, and optical fields), the dynamics of droplets or bubbles can be accordingly controlled and their motions are subsequently directed (7–11). For example, magnetic and electric actuation require magneto-responsive and dielectric surfaces, but the need to manufacture patterning electrodes on substrate or embedding magnetic particles into a soft matrix increases the complexity of the overall operation (7, 8). Acoustic method offers the extraordinary capacity to trap and manipulate bubbles by forming the standing waves, but the precise control of bubble location in three dimensions still remains challenging (9, 10). Regarded as one of the promising methods for the remote and contactless manipulation, the optical approach such as the optical tweezers can trap high-refractive-index objects such as particles, but low-refractive-index objects such as bubbles are anti-trapped (36).”

The claim for industrial relevance (p13 and introduction) also seems difficult to support, since the technique applies to the manipulation of single bubbles/droplets.

Thanks for the comment! It is true that the current setup is difficult to support the industrial applications we referred to, but the proposed technique or method opens up a new possible way or sheds light on the relevant applications. For example, using bubbles for wastewater treatments or microorganism removal has been proven as a

sustainable and environmentally friendly method in many industrial applications [T. Temesgen, Micro and nanobubble technologies as a new horizon for water-treatment techniques: A review, Adv. Colloid Interface Sci. 246, 40 (2017)].

Also, from a mechanics point of view, the shear stress when bubble impacting and sliding on a wall plays an important role in removing biofilms and contamination from various surfaces [Esmaili, Bubble impact on a tilted wall: Removing bacteria using bubbles, Phys. Rev. Fluid 4, 043603 (2019)]. Therefore, the laser manipulation of underwater bubble bouncing and moving is well suited for surface cleaning.

For material fabrication, the three-dimensional non-contact manipulation of underwater bubbles can induce bubbles to interact with other suspended or stationary bubbles/droplets for the construction of "gas-in-liquid" or "liquid-in-gas" composite capsules, such as the NIR-laser triggered core coalescence of double-emulsion drops.

Finally, since the Marangoni convection around thermal bubbles can drive the surrounding fluid to generate vortex patterns, which efficiently capture and enrich microparticles on the bubble surface for carrying and transferring drug molecules, cells, microorganisms. This particle enrichment and 3D manipulation of the bubble might be relevant for applications such as wastewater treatment or targeted drug delivery.

Actually, the technique we propose here is not only applicable for manipulation of single bubble/droplet, but also possibly feasible for multiple bubbles. The single laser beam can be easily split into several beams or programmed using a beam splitting element (diffractive optical elements) or spatial light modulators. Thus, generation and manipulation of multi-bubbles simultaneously can be achieved, likely holding promising for the industrial application.

Finally, oscillating bubbles induced by (thermal) Maragoni forces has been the object of several recent papers which is a concern for Nature Communications. Nevertheless, I do see the uniqueness of this particular physical configuration and I believe the novelty claims should be best focused on how this effect is put to use in the third part of the paper by further detailing the unique assemblies this technique can offer.

Thanks for the comment!

The main originality or novelty of this work is briefly summarized as following:

- (1) We observe the vertical bouncing bubble during laser impacting on water.
- (2) We propose the underlying physical mechanism based on the unique temperature inversion layer, which is related with the high thermal conductivity of the cover glass.
- (3) We show the bouncing bubble can be steered horizontally by the laser, resulting in the dancing bubble.
- (4) We perform the scaling law analysis and numerical simulation, and the theory has a remarkable agreement with experiments.

In terms of the potential applications, previous work on bubble manipulation by laser is focused on bubble following the moving laser spot or bubble trapped by a focused laser spot. Here, by designing the temperature field along laser propagation direction, we have successfully realized predictable bubble bouncing behavior consequently extending a new dimension (along the propagation direction) for bubble manipulation by laser.

Fig. 5 Steering the bubble to interplay with bubbles, droplets and particles. (A) 3D manipulation of the dancing bubble, as shown by its trajectory with the translation speed of 2 mm/s, the red arrow for the initial position. (B) Numerically demonstration for crossing obstacles of the dancing bubble. (C) The coalescence of bubbles (b and c) with the dancing bubble (a) along its pathway under a moving speed of 5 mm/s. (D) The encapsulation of the dancing bubble by a dyed hexane droplet from the side and top view, producing a core-shell structure of bubble-droplet composite. (E) Particle collection unto the surface of the dancing bubble during its sweeping through the water, top and side view for PS beads (diameter of 80 μm) collected by the bubble. Scale bar: 200 μm

This dancing bubble in our work has two remarkable features, one is bouncing, and the other is steerability (Fig .5). Fig5A and Fig 5B demonstrate these two features simultaneously, allowing 3D motion and leaping over a wall. Fig 5C-E mainly show its steerability to interact with the preexisting objects including walls, bubbles, droplets or nanoparticles, although bouncing might not be necessarily related.

Nevertheless, due to the extended moving dimension of bouncing bubble and the Marangoni convection around the bubble, the dancing bubble has a unique advantage in particle collection and transportation (Fig 5E) taken as an example, hence in a fashion similar to “deep surface cleaning”, particles can be collected or trapped by the bubble within a much larger volume (related with the vertical bouncing and horizontal translation). This novel capability of particle collection and transportation is demonstrated in the revised Supplementary Movie S10. Clearly, this remarkable particle enrichment and 3D manipulation of the bubble has implications for applications such as wastewater treatment or targeted drug delivery. Compared with other methods to trap particles, such as acoustically activated bubbles or previous optical manipulated bubbles, the underwater spherical dancing bubble in our system shows an obvious superiority in terms of enrichment concentration and flexible trajectory.

other comments/concerns:

1. *“the observed 1/2 scaling relation of temperature evolution”. I am puzzled by this statement. First, based on the supplementary information (Fig. S3B), the scaling in 1/2 is not very convincing. In addition, 1/2 for heat diffusion (disregarding flow during transient build-up) corresponds to a plane, 1D cartesian diffusion. In Fig. S3B, we can see clearly the 1 power law for short time, corresponding to a (mostly) spherical diffusion of the heat deposited in the bulk by the gaussian Beam. In this context, isn't the decrease of slope observed after 1 second simply the slow transition toward the steady state temperature profile (slope = 0) existing for spherical heat diffusion? (Note that 1 s would be consistent with the typical diffusion timescale given the size of the laser beam). In this case, the effective power law would depend on the time at which nucleation occurs.*

If this is not the case, why does the power law in Fig S3B differs so significantly from 1/2 and why would the authors expect 1/2?.

Finally, this scaling is bound to lose validity once the bubble is formed since it acts as a heat sink and changes the local absorption and light transmission. It is thus not trivial to me, based on the proposed arguments, why one would still expect a linear bubble growth over time.

Thanks for the comment. For a slender cylindrical laser beam with a steady power irradiating water, the temperature evolution has two limits: $\Delta T \propto Pt$ for short time period by ignoring thermal diffusion (Fig. S4B and Fig. S4C), while $\Delta T \propto Pr_l^2/k$ for long time period by considering the balance of laser heating flux and thermal conductive flux.

Fig. S4 Measured temperature during laser impacting on water. (A) Evolution of peak temperature (T_{peak}) and the corresponding location δ_{inv} ($P=15$ W with bubble formation) from the thermal images. (B, C) During the initial pre-heating stage, the evolution of elevated temperature (ΔT) for $P=15$ W with and $P=10$ W without bubble formation.

Actually, in previous supplementary materials, the long-time temperature evolution is used to explain the almost linear growth of bubble radius, but it does not affect other analysis for bubble motion. In the revision, in order to better focus on the bouncing behavior in this work, the analysis of 1/2 scaling of long-time temperature evolution and the linear growth of bubble are removed from the main text and supplementary materials in the revision, and the explanation of bubble motion is not affected.

2. P4: “The existence of temperature inversion layer is attributed to the viscous and thermal boundary layers (BLs), as swept by the thermally-driven buoyancy flow during laser heating”. What is dominant: the flow or the large conductivity of the sapphire that thus evacuates much heat near its surface?

Thanks for this wonderful comment! We have clarified the underlying physical mechanism of temperature inversion layer in the revision.

In short, thermal boundary layer (δ_{th}) is caused by the thermally-driven buoyancy flow. The temperature inversion layer indicates the temperature distribution is non-monotonic, and its thickness (δ_{inv}) is determined by determined by the competition between laser heating rate and cover cooling rate. More quantitatively, δ_{inv}/δ_{th} depends on the ratio of their thermal effusivity between cover materials and water, as shown below (the same as Figure S6C in Supplementary).

Figure S6C: The thickness of temperature inversion layer dependent on the thermal effusivity of cover material

Below we will clarify these points, particularly the effect of thermal conductivity of cover materials on δ_{inv} .

- I. For the model for the velocity and thermal boundary layer (δ_{th}), detail derivation is provided in the revised supplementary Section S6 “Model for velocity and thermal boundary layer”.

To model the boundary layer induced by the buoyancy flow sweeping the cover surface, we first consider the buoyancy flow similar to the analysis for laminar natural convection on a heated vertical surface [*Fundamentals of Heat and Mass Transfer (John Wiley & Sons, 2011)*], which is an axisymmetric model. The expression of buoyancy flow velocity is obtained $V_b = \sqrt{g\beta\Delta TL} = \frac{\mu}{\rho L} Gr^{1/2}$. Then, we use the hydrodynamic solution by Blasius for laminar flow over isothermal plate [*Fundamentals of Heat and Mass Transfer (John Wiley & Sons, 2011)*] to obtain the expression of velocity and thermal boundary layer thickness.

- II. We have carefully elaborated the effect of the thermal conductivity of the cover materials by the following revisions:

- Perform experiments for various cover materials, as shown in Supplementary Section 3 “The effect of cover materials on the bouncing behavior” with Fig. S3;
- Build model to understand temperature inversion layer, as shown in Supplementary Section 7 “Model for temperature inversion layer” with Fig. S6;
- Clarify its physical mechanism in the section of “Formation of temperature inversion layer” in the main text.

Table S1: Material Properties ($T = 293.15\text{K}$)

Properties	Density	Thermal conductivity	Heat capacity	Thermal diffusivity	Thermal effusivity
Notation	ρ	k	c_p	κ	$e_i = \sqrt{k_i \rho_i c_{pi}}$
Units	kg/m^3	$\text{W}/(\text{m} \cdot \text{K})$	$\text{kJ}/(\text{kg} \cdot \text{K})$	m^2/s	$\text{W}\sqrt{\text{s}}/\text{m}^2\text{K}$
Water	1000	0.59	4.2	1.4×10^{-7}	1574.2
Sapphire	3980	23.1	0.761	7.63×10^{-6}	8364.5
Quartz	2200	1.32	0.772	7.77×10^{-7}	1497.3
PMMA	1190	0.19	1.42	1.12×10^{-7}	566.6
PDMS	970	0.16	1.46	1.13×10^{-7}	476.0

Figure S3: Snapshots and comparison for experiments with different cover materials. (A) High-speed images for the produced bubble within water for PMMA cover. (B-D) The bubble radius R , central position H_c , and topmost position H_t versus time t .

In experiments, to further verify the effect of thermal conductivity of the top glass on the formation of temperature inversion layer, we tested various glasses with different thermal conductivity (see Table S1). The bubble bouncing behavior still holds for higher thermal conductivity cover such as quartz and sapphire cover, while bouncing disappears for lower thermal conductivity cover, such as PMMA and PDMS cover. As shown in Figure S3, no bouncing is observed for the PMMA cover, and bubble just hangs at the solid-liquid interface.

In simulation, the corresponding simulated temperature distribution shown in Fig. S6A also indicates that the thickness of temperature inversion layer becomes thinner for cover materials with a lower thermal effusivity, thus the temperature inversion layer is too thin to cause bubble bouncing for the case of PMMA and PDMS.

In theory, to qualitatively understand temperature inversion layer, we build the model to demonstrate the relevant physical parameters for the thickness of temperature inversion layer. The dependence of ratio between the thickness of temperature inversion layer (δ_{inv}) and thickness of thermal boundary layer (δ_{th}) on the ratio between thermal effusivity of cover material (e_s) and that of water (e_w) is shown in Fig. S6C. The detail derivation is provided in the revised supplementary Section 7 “Model for temperature inversion layer”.

In the limit of high effusivity or thermal conductivity (such as sapphire), the thickness of temperature inversion layer nearly approaches that of the thermal boundary layer ($\delta_{inv} \approx \delta_{th}$). Then, the thickness of temperature inversion layer extracted from simulation agree well with the scaling relation $\delta \propto P^{-1/4}$ for the analysis of the thickness of thermal boundary layer (Fig. S6D).

Figure S6: Model for TIL. (A) Temperature profile for different cover materials obtained from simulation. (B) Temperature profile for sapphire cover with different thickness obtained from simulation. (C) Dimensionless thickness of TIL dependent on the thermal effusivity of cover material. (D) The scaling relation between the thickness of TIL and laser power obtained from simulation.

3. *Thermal imaging: the authors do not provide much information on the thermal imaging. How do the authors solve the issue of imaging quantitatively through a significant layer of water? In particular (p15), how is the thermal imaging calibrated?*

Thanks for the nice comment! More details of the temperature measurements are provided here. In order to gather accurate temperature measurements, the temperature detection of the superheated liquid from the side view is calibrated, since the side wall of the quartz cuvette affects the infrared intensity signal to be detected by the camera from the target liquid surface.

Firstly, the radiometric thermal camera is so close to the target surface (less than 20 cm) that the atmospheric transmission factors can be neglected. Secondly, since the surface emissivity of both the quartz glass ($\epsilon=0.93$) and liquid water ($\epsilon=0.96$) is large

(greater than 0.90), the impact of background temperature reflection can also be ignored. Thirdly, we calibrate the measured temperature from the IR camera by using a correction factor. The factor is obtained by simultaneously using the thermal camera and a thermocouple to measure the temperature evolution of a cooling process of hot glycerol ($\epsilon=0.96$, the same emissivity as that of water) in the cuvette within the temperature range from 140 °C to 30 °C. And the location of the thermocouple is exactly the same with the laser focus spot, which is very near to the quartz wall (about several millimeter). Hence, the calibrated temperature refers to the part of the irradiated liquid in the plane of laser transmission.

A new section of “Temperature measurement” is added in Materials and Methods in the main text. “An infrared thermal camera (FLIR A6750sc) was used to acquire the temperature evolution of liquid during laser irradiating. For the top view temperature detection, the sapphire glass with high transmissivity @185-5000 nm was adopted as the cover glass since infrared light can optically penetrate the sapphire glass. Since the side wall of the quartz cuvette affects the detection of infrared intensity signal emitted from the target liquid surface, the measured temperature values from the side view by the thermal camera was required to be calibrated. For the calibration, a thermocouple was exactly located at the same position of laser focus spot, closer to the quartz wall (about several millimeter). By using the thermal camera and the thermocouple to simultaneously measure the temperature evolution of a cooling process for a glycerol in the cuvette within the temperature range from 140 °C to 30 °C (the same emissivity with that of water, $\epsilon = 0.96$), the elevated temperature of the irradiated liquid in the plane of laser transmission was quantitatively calibrated.”

4. *When the bubble is partly in and partly outside the boundary layer, can one simply consider 2 opposite forces, where the Marangoni force induced by the boundary layer is stronger (as suggested in the article) or does the bubble behaves as a dipole source for the flow? What would this difference entail?*

Thanks for the nice comment! When some part of the bubble in the temperature inversion layer and the other part in the normal temperature gradient, there exists a competition of buoyancy force F_b , upward Marangoni force F_m^+ and downward Marangoni force F_m^- . Hence, in the revised Fig. 3D in the revised manuscript, we carefully compare the magnitude of these three forces: buoyancy force, upward thermal Marangoni force and downward Marangoni force.

For the case that bubble touches surface (as shown in Fig. 3C), from the view of scaling, the magnitude of the downward Marangoni force $F_m^- = \Delta\gamma \cdot R = \frac{d\gamma}{dT} \frac{dT^-}{dz} \cdot \min(2R, \delta_{inv}) \cdot \pi R$, where dT^-/dz for the temperature gradient within temperature inversion layer is about 30 K/mm. The magnitude of the upward Marangoni force $F_m^+ = \Delta\gamma \cdot R = \frac{d\gamma}{dT} \frac{dT^+}{dz} \cdot \max(0, 2R - \delta_{inv}) \cdot \pi R$, where dT^+/dz for the temperature gradient outside temperature inversion layer is about 1 K/mm. Thus, when the bubble touches

surface, the downward Marangoni force is always at least one order larger than the upward Marangoni force. Therefore, to simplify the analysis, the upward Marangoni force can be reasonably neglected.

In the revised manuscript, we have presented the magnitude of the upward Marangoni force (F_m^+) in Fig. 3D to compare the individual component clearly.

Fig. 3D Sketch of the competing buoyancy force and the thermal Marangoni force, and their magnitudes dependent on bubble radius

5. P2: “By utilizing an acoustic or magnetic field, non-invasive manipulation of bubbles is achieved, but sophisticated systems or potentially harmful sample pretreatments for additional properties are required” This is not necessarily true for acoustics- or optics-based techniques.

Thanks for the comment! We have refined the expression in Introduction on page 2, “Other possible approaches utilize the external stimuli to actively drive the motion of droplets or bubbles, including magnetic, electric, acoustic or light fields (7–11), and their dynamics is correspondingly restricted by the applied specific forces”.

We have included the comparison of these technologies in Discussion on page 14, “Conventionally, by utilizing the applied external stimuli (such as magnetic, electric, acoustic, and optical fields), the dynamics of droplets or bubbles can be accordingly controlled and their motions are subsequently directed (7–11). For example, magnetic and electric actuation require magneto-responsive and dielectric surfaces, but the need to manufacture patterning electrodes on substrate or embedding magnetic particles into a soft matrix increases the complexity of the overall operation (7, 8). Acoustic method offers the extraordinary capacity to trap and manipulate bubbles by forming the standing waves, but the precise control of bubble location in three dimensions still remains challenging (9, 10). Regarded as one of the promising methods for the remote and contactless manipulation, the optical approach such as the optical tweezers can trap high-refractive-index objects such as particles, but low-refractive-index objects such as bubbles are anti-trapped (36).”

6. *Are the oscillations in Fig 1B real or only an effect of imperfect detection? If they are real, what is their source?*

Thanks for the comment! The oscillations of bubble radius in Fig. 1B can be divided into two kinds: one is the increased steps, which is caused by the limitation of spatial resolution, the other is strong oscillations during bubble bouncing, which may be caused by the bubble deformation and the inner pressure fluctuation varied with the surrounding liquid temperature. We identify the bubble bouncing based on the time-dependent position of bubble center (H_c as shown in Fig. 1C).

7. *P5: “the temperature peak is confirmed to be around hundreds of micrometers below the interface”. Experimentally, the inversion goes until 0.4mm, this is twice as much as in the simulation. Furthermore, the profiles differ. Where do these differences come from? Finally, the experimental curve is plot in semilog and the simulation curve has linear axes. This should be uniform. Is the temperature profile decay for large z comparable in both cases?*

Thanks for the comment! In the revised manuscript, we uniform the x axis (Fig. 2B and E) in linear scale, and the simulated temperature data for sapphire with thickness of 1 mm (blue line in previous Fig. 2E) is replaced by data for sapphire with thickness of 250 μm (blue line in revised Fig. 2E) which is consistent with the glass thickness (250 μm) in experiment.

The thickness of temperature inversion layer (δ_{inv}) is fluctuating during the bouncing process but has an average value of about 0.3 mm, as shown in Fig. S3A. In the revised Fig. 2B and E, the thickness of temperature inversion layer is about 0.3 mm experimentally and numerically, also their profiles are similar.

Since temperature profile is simulated to validate the existence of the temperature inversion layer due to the high thermal conductivity of the cover glass, temperature profile decays for large Z is not focused here.

8. *Fig 2E: PMMA has a lower conductivity than water. If, as the abstract suggests, thermal conductivity is key, why does it still induce a temperature inversion (albeit a small one)?*

Thanks for the comment. As addressed in Question 2 together with Fig S6C.

9. *Equation 4: although the scaling obtained seems to work, I am confused by the approach: the instability and thus the oscillation frequency in both cases originate from the competition of 2 opposing forces. Without either one, there would be no frequency. In term of oscillator, it is therefore the dependence on z of the resulting force that gives rise to the existence of an eigen frequency. How correct is then this analysis? This scaling implicitly considers both forces to be equal and opposite. I think these considerations should be more clearly introduced and discussed.*

Thanks for the comment. The bouncing bubble system is more like the parabolic motion rather than the oscillator. The downward Marangoni force induced by temperature inversion layer provides the initial velocity of bubble motion. Then, the motion of bubble destructs the temperature inversion layer. The buoyancy force and upward Marangoni force (induced by the Beer-Lambert law) act as the restoring forces, then the bubble returns back to the cover, and ends a cycle of bounce.

Thus, the oscillation frequency in our experiments is obtained not from eigen frequency, but from solving the bubble motion equation. This idea based on the bubble motion equation is similar to study bouncing bubble induced by the competition between driving solute Marangoni force and restoring thermal Marangoni force in a recent literature [Zeng, et al, Proc. Natl. Acad. Sci. U. S. A. 118 (2021)]. Furthermore, the detail derivation process is provided in the revised Supplementary Section 8.

Also we compare the estimated value in analysis and the observed value in experiments. For small bubble, the upward Marangoni force F_m^+ is dominant, and the obtained expression of frequency $f = C_1 R^{-1/2}$. The estimated value of the prefactor $C_1 \approx 10$, and the observed value of the prefactor $C_1 \approx 16$ in Fig. 3F. For large bubble, the buoyancy force F_b is dominant. For bubble radius $R = 0.9$ mm, the estimated frequency $f = 19.6$ Hz, and the observed frequency is $f = 19.2 \pm 2.17$ Hz in Fig. 3F.

10. P11: “the obtained maximum translation speed is consistent with the experimental observation, which is about 40 mm/s” How does this maximum translation speed depend on the laser power?

Thanks for the comment! From the view of Eq. S32 in the revised Supplementary Section 9, the maximum translation speed $v_{cr} = \sqrt{\left| \frac{d\gamma}{dT} \right| \cdot \frac{\alpha\eta(z)P}{3\pi\rho c_p\mu} \cdot \frac{r_b\Delta l^2}{r_l^4} e^{-\Delta l^2/r_l^2}}$, which is proportional to $P^{1/2}$. In our experiment, since the threshold of laser power for bubble nucleation is about 15 W, and the maximum output laser power of the laser source is limited to 30 W. This limited range of available laser power in experiments consequently hinders the quantitative data to investigate the relationship between the maximum translation speed and the laser power experimentally.

11. P13: “the pre-existing bubble b and c along the pathway at 0.8 s, resulting in the sequential capture and fast coalescence at 0.9 and 1.0 s. Then the combined large bubble a + b + c continues its motion as guided by the laser (Movie S7).” I do not see the practical interest of this: there are many ways to induce coalescence, the difficulty is to prevent it.

Thanks for the comment! The dancing bubble reported in this work has two remarkable features, one is bouncing, and the other is steerability. In the revised Fig 5, Fig5A and Fig 5B demonstrate these two features simultaneously, allowing 3D motion and leaping over a wall. Fig 5C-E mainly show its steerability to interact with the preexisting objects including walls, bubbles, droplets or nanoparticles.

To prevent bubble coalescence, our method to obtain dancing bubble might be relevant by jumping over the pre-existing bubble, but requires more future work.

12. P14: *“the solutal Marangoni force is vanished, and a distinct underlying mechanism for the bouncing motion is attributed to the thermal Marangoni”* This statement is confusing: from the title of the reference discussed, it appears to also consider thermal Maragoni as (partial) driving mechanism.

Thanks for the comment! In this reference, they use solute Marangoni force as a driving force, and thermal Marangoni force as a restoring force, which is different from the situation in our manuscript that using bidirectional thermal Marangoni force and buoyancy force to achieve bouncing.

To clarify the unnecessary confusion, the sentence has been modified in the main text on page 15 as below, “We note that in the recent work in Ref (23), bouncing plasmonic bubble has been demonstrated in a binary liquid consisting of water and ethanol, and the competition between the solutal and thermal Marangoni forces is identified as the origin of the periodic bouncing. However, with only pure water as the host fluid here, both the bidirectional thermal Marangoni force and buoyancy force are implemented carefully to achieve bouncing.”.

13. P16: section *“PIV measurement for thermal buoyancy flow induced by laser heating”*. Since the PIV measurements are only shown in SI, I do not think it should be in the methods of the article itself.

Thanks for suggestion! This part has been removed in the methods and included in the Supplementary Section 6 in the revised manuscript.

14. Fig 2D: *the temperature profile does not seem to match Fig 2A. Why is that?*

Thanks for the comment. As limited by the spatial resolution of thermal camera, the snapshots from thermal camera is used to show the overall distribution. The simulation snapshot is a zoom-in one to show more detail within temperature inversion layer. The differences in experimental and numerical temperature profile might be due to the effect of bubble formation and bounce.

15. *The x-axes in Fig 2 B and D should have the same limits. Furthermore, these plots appear never to reach the state where the buoyancy force overcomes the Maragoni force.*

Thanks for the comment! We have updated the corresponding figure accordingly (Fig. 3B and D). When the bubble radius exceeds R_{up} , the buoyancy force F_b overcomes the downward Marangoni force F_m^- , and the bubble cannot bounce any more.

Fig. 3 B and D Sketch of the competing buoyancy force and the thermal Marangoni force, and their magnitudes dependent on bubble radius.

16. Fig 2D: why does the F_m^- force keep increasing after $\delta_{inv}/2$, while the bubble now experiences a opposite Maragoni effect?

Thanks for the nice comment! When some part of the bubble in the temperature inversion layer and the other part in the normal temperature gradient, there exists a competition of buoyancy force F_b , upward Marangoni force F_m^+ and downward Marangoni force F_m^- . Hence, in the revised Fig. 3D in the revised manuscript, we carefully compare the magnitude of these three forces: buoyancy force, upward thermal Marangoni force and downward Marangoni force.

For the case that bubble touches surface (as shown in Fig. 3C), from the view of scaling, the magnitude of the downward Marangoni force $F_m^- = \Delta\gamma \cdot R = \frac{d\gamma}{dT} \frac{dT^-}{dz} \cdot \min(2R, \delta_{inv}) \cdot \pi R$, where dT^-/dz for the temperature gradient within temperature inversion layer is about 30 K/mm. The magnitude of the upward Marangoni force $F_m^+ = \Delta\gamma \cdot R = \frac{d\gamma}{dT} \frac{dT^+}{dz} \cdot \max(0, 2R - \delta_{inv}) \cdot \pi R$, where dT^+/dz for the temperature gradient outside temperature inversion layer is about 1 K/mm. Thus, when the bubble touches surface, the downward Marangoni force is always at least one order larger than the upward Marangoni force. Therefore, to simplify the analysis, the upward Marangoni force can be reasonably neglected.

In the revised manuscript, we have presented the magnitude of the upward Marangoni force (F_m^+) in Fig. 3D to compare the individual component clearly.

Fig. 3D Sketch of the competing buoyancy force and the thermal Marangoni force, and their magnitudes dependent on bubble radius

Minor comments:

Fig 2B: is the substrate interface at $z = 0$?

Yes! We have added this definition, “ $Z = 0$ refers to the glass/water interface” in Fig 2 caption in the main text.

Which laser power was used to calculate Fig 3 B and D?

We have included the laser power in the caption of Fig. 3 ($P = 20$ W) .

It would be helpful if R_{up} and R_{low} are depicted in Fig 3 B and D.

Thanks! We have depicted R_{up} in Fig. 3D, but R_{low} is not from the force analysis.

Eq. 2: where do these scaling laws exactly come from.

In the revised manuscript, the full expressions of F_m^+ and F_m^- are given. The downward Marangoni force $F_m^- = \Delta\gamma \cdot R = \frac{d\gamma}{dT} \frac{dT^-}{dz} \cdot \min(2R, \delta_{inv}) \cdot \pi R$, and the upward Marangoni force $F_m^+ = \Delta\gamma \cdot R = \frac{d\gamma}{dT} \frac{dT^+}{dz} \cdot 2\pi R^2$. The detail derivation process of analysis is provided in the revised Supplementary Section 5-9.

P8, last line. The scaling for the thermal boundary layer thickness is obtained from this is a dimensional analysis with 3 nondimensional numbers. However, it entails non-trivial interconnected effects. Since the authors have thermal imaging capability, it would be interesting to verify this law experimentally.

Furthermore, why would this law remain true in the presence of a bubble whose presence changes heat deposition and induces its own mixing?

Details of the scaling for the thermal boundary layer thickness can be seen in the revised supplementary section 6 (“Model for velocity and thermal boundary layer”).

To model the boundary layer induced by the buoyancy flow sweeping the cover surface, we first consider the buoyancy flow similar to the analysis for laminar natural convection on a heated vertical surface [*Fundamentals of Heat and Mass Transfer (John Wiley & Sons, 2011)*], which is an axisymmetric model. The expression of buoyancy flow velocity is obtained $V_b = \sqrt{g\beta\Delta TL} = \frac{\mu}{\rho L} Gr^{1/2}$. Then, we use the hydrodynamic solution by Blasius for laminar flow over isothermal plate [*Fundamentals of Heat and Mass Transfer (John Wiley & Sons, 2011)*] to obtain the expression of velocity and thermal boundary layer thickness. Then, the thickness of velocity boundary layer $\delta_v = 5L/\sqrt{Re}$, and the thickness of thermal boundary layer $\delta_{th} = \delta_v/Pr^{1/3}$. The physical relation between the thickness of temperature inversion

layer (δ_{inv}) and that of thermal boundary layer (δ_{th}) has been clarified in the revised Supplementary Section 7.

In fact, the formation of temperature inversion layer in our experiments is associated with laser heating effect and stable convection fluid flow (about 0.01 m/s in experiments), while the flow induced by bubble motion (~ 0.1 m/s) is one order larger than the buoyancy flow. Then, the temperature inversion layer actually experiences a construction-destruction-reconstruction process during bubble motion. This complicated dynamical process hinders to verify the formula experimentally.

P9: “by setting $H_{b;cr} = 0.1R$.” What is the physical argument behind this operation?

Thanks for the comment. As shown in Fig. 1C, we note that the bouncing begins with a small amplitude, to identify the lower threshold of bubble bouncing, a criterion for the bubble vertical displacement H_t is needed. Thus, to eliminate the effect of fluctuation of bubble radius and lateral oscillation of bubble, we set the low bouncing threshold of $H_{t,cr} \approx 0.1R_{low}$ that only when the bubble vertical displacement is comparable with 0.1 of its radius, based on the argument that bouncing becomes observable by high-speed images feasibly. The detail derivation is provided in the revised Supplementary Section 8.

Eq.4: why keeping the pi in the dimensional analysis when the 4/3 is already removed?

We have removed this coefficient accordingly.

Eq. 4: Equal sign of approximately equal?

We have changed equal sign “ \sim ” to “ \propto ” accordingly.

Text:

“followability”: to me, this appear not to be the best choice of word here since the authors discuss the capacity of the bubble to follow the laser rather than the capacity of the laser to be followed.

The word “followability” is replaced by “steerability” in the revised manuscript.

P2: ‘fascinating’. I would suggest letting the reader judge of this.

We have removed the word “fascinating”.

P3: “Here based on the technological advancement of laser impacting on water, by simply designing a specific thermally conductive interface, we directly realize both the vertical bouncing and the horizontal translating of the millimeter-sized bubble within pure water, achieving the dancing bubble within water in 3D.” Please rephrase.

The sentence has been rephrased as below “Here, by adopting a near-infrared laser irradiating into pure water through a transparent glass cover with a high thermal conductivity, we observe a vertical bouncing behavior of an optothermal bubble...Moreover, being subject to the navigation of the laser beam, the resulted dancing bubble can be translated along the horizontal direction.”

P3: “Additionally, we demonstrate the remarkable manipulation capability of the bubble to interact with the preexisted droplet or nanoparticles” Please rephrase.

The sentence has been rephrased as below “Further, through the precise control of the specific interaction with the preexisting objects, we successfully demonstrate the unique bouncing and steerability features of this dancing bubble”.

P4: “During the whole operation process around 50 seconds,” please rephrase

The sentence has been rephrased as below “During the bubble growth within 50 s”

P4: “robust or resilient” Do the authors mean “robust and resilient”?

This sentence has been removed in the revised manuscript.

P4: “at $t = t_0$ ” the plots refer to $t=0$.

We define $t=0$ as the moment right before the bubble is visible, and $t = t_0 = 18.48$ s in Fig. 2A and 2B refers to the onset moment of one bouncing cycle. The sentence “Fig. 2A and B at $t=t_0$ and $t_0 + 56$ ms” has been removed in the revision accordingly.

P6: “Such the temperature inversion might be responsible for the subsequent bouncing bubble.” Please rephrase.

This sentence has been rephrased as below, “Particularly, for the glass cover with a high thermal conductivity, within TIL, the temperature gradient up to 10 K/mm (Fig. S4) might be responsible for the bouncing behavior.”

P8: “two important issues are required to be addressed:” please rephrase

Thanks! This sentence has been removed in the revised main text.

P9: “converted into kinetic energy of bubble with the initial velocity” please rephrase

This sentence has been rephrased as below “From energy perspective, the work done by the downward Marangoni force ($W_m^- \propto \gamma_{th}^- R^2 \delta_{inv}$) is mainly converted into the kinetic energy of bubble associated with an initial velocity ($v_{b,0}$), $E_k \propto \rho R^3 v_{b,0}^2$.”

P9: “distance an be evaluated” -> “can”

This typo has been corrected --- “distance can be evaluated”.

P11: “The followability of bubble with laser is driven” please rephrase

This sentence has been rephrased as below “The horizontal translation of the bubble is driven by the thermal Marangoni force”.

Caption Fig.5: “Steering the bubble to interplay with the bubble, droplet and particles”. Please make the plurals consistent.

Thanks! In the revised main text, this sentence has been replaced by “Utilizing the features of the dancing bubble” in the caption of Fig. 5.

P13: “This unique capability of 3D manipulation for the bubble demonstrates the potential for the versatile applications” please rephrase.

Thanks! In the revised main text, this sentence has been replaced by “this spontaneous dancing behavior together with the horizontal motion under navigation of the laser allows the attainment of complex movement in three dimensions, as indicated by its prescribed well-defined trajectory of the cross shape”.

P14: “Marangoni force is vanished,”

Thanks! This paragraph has been rephrased as below “We note that in the recent work in Ref (23), bouncing plasmonic bubble has been demonstrated in a binary liquid consisting of water and ethanol, and the competition between the solutal and thermal Marangoni forces is identified as the origin of the periodic bouncing. However, with only pure water as the host fluid here, both the bidirectional thermal Marangoni force and buoyancy force are implemented carefully to achieve bouncing.”.

P14: “of bubble are different in many orders of magnitude”

Thanks! This sentence has been rephrased as below “Also, the bouncing frequency and the size of bubble are different in orders of magnitude for both works”.

P17: “considering interfacial tension as a linear function with temperature”

This sentence has been rephrased as below “The surface tension is assumed as a linear function of temperature $\gamma(T) = \gamma_0 + \frac{d\gamma}{dT}(T - T_0)$ ”.

REVIEWERS' COMMENTS

Reviewer #1 (Remarks to the Author):

In the revised manuscript, the authors conducted additional experiments with different cover materials. The underlying mechanism is further verified. They also provided detailed analysis of different kinds of forces experienced by a bouncing bubble. Overall, they properly replied to my comments. Here I only have three minor comments before the manuscript can be accepted.

1. Throughout the main manuscript and supplementary materials, different values of temperature gradient were used. For example, the authors used 10 K/mm on page 4 and 6. On page 18, they used 50 K/mm. In the supplementary materials, they used 30 K/mm (page 18) and 100 K/mm (page 21). Authors should make the value consistent throughout the manuscript. Otherwise, it confuses readers when it comes to different values.
2. Please check if the expression of the upward Marangoni force in Eq. S21 on page 20 of the supplementary materials is correct. Authors used the upward Marangoni force expression corresponding to Eq. S17, which is the case when a bubble completely moves outside TIL. Since H_t is within 0.1 of bubble radius, I assume the bubble is mostly immersed in TIL. Then Eq. S16 other than Eq. S17 is applicable. Can authors clarify this?
3. In Fig.S9, does the temperature gradient outside TIL need to be zero? If not, please correct it.

Reviewer #2 (Remarks to the Author):

I thank the authors for their detailed answers and the many clarifications they have brought to both the paper and the supplementary materials. They have clarified the major concerns I had. Their analysis and discussions are, now, presented in a clear, refutable and convincing way. I think the work they present is worth being published.

I think they should still adjust one explanation regarding point 1 of my previous report, which does not change the following of their analysis. The bouncing phenomenon is observed for the 'long-time' regime of the heating of the liquid by the laser. Therefore, equations S3 and S4 are not relevant. They should rather be replaced by the steady, radial heat equation $(1/r)d(rk dT/dr)/dr + A0P \exp(-r^2/r_l^2) = 0$ and its solution, which is also linear in P.

This also suggests that the authors should simply remove all detailed discussions regarding the short-time (unsteady) regime when the liquid temperature is increasing. This is unnecessary and may bring confusion.

Minor: the prefactor in (S15) is not correct, it should be 6π for a solid particle and less (depending on surface condition) for a bubble.

Reviewer #3 (Remarks to the Author):

I thank the authors for carefully considering and addressing my comments. The answers are mostly satisfactory. There are only a few aspects with which I am not fully satisfied. I recommend conditional acceptance, pending the last few revisions.

>> The authors have written a very interesting discussion on alternative technologies. However, I perceive the discussion as flawed in some respects. Without into the details of

specific implementations, both optical and acoustical tweezer have been used to manipulate bubbles. Although I have not used the techniques myself, optical tweezers seem suitable for 3D bubble manipulation without “complex” grating by adapting the laser profile (see [1]). Likewise, SAW is a specific implementation, but acoustic tweezers can also be conceived with transducer arrays, which is a fairly common technology. I do recognize the size-dependent direction of the force as a difficulty.

In short, acoustical and optical tweezer can be used for 3D bubble manipulation. The technique presented by the authors allows mainly 2D manipulation with the constrain of a highly conduction interface in immediate vicinity. The discussion proposed thus need to be revised to account for (1) the fact that the proposed limitation for acoustical tweezers only apply on SAW-based systems (unless shown otherwise), (2) that the limitation on the optical tweezer is thus not justified, and (3) the fact that the discussion does not explicitly explain the added value of the new methods (even if the added value is potentially found for specific cases). I think a clear and solid discussion on the fundamental point is necessary for the broad audience of Nature communications.

>> This are convincing example. Can the authors maybe make use of the half line left in the introduction to explicitly mention the most relevant applications?

>> For the uniqueness:

(1) Bouncing of a bubble due to laser irradiation was also observed in [2] (cited by the authors)

(2-4) I agree.

I also concur with the rest of the argumentation. In hindsight, I think it is precisely point (1) that triggered me since the bouncing itself is not new, but the detail physical mechanism is. I would encourage the authors to point out this distinction. Last point with respect to the “obvious superiority”, I have yet to see any proof in that regard as this is not trivial to me. Maybe a topic for a follow-up investigation.

1. I am confused by the argument. First, the consideration of a rather cylindrical geometry or spherical symmetry depends on the attenuation of the laser as it is absorbed. I do not have the information to know which configuration is appropriate. However, a strictly cylindrical problem requires infinite power (or at least a heating power per unit length) and does not have a steady state. A non-diverging solution can only exist for a finite size source in 3D space. Nevertheless, removing the problematic discussion from the text is fine with me.

2. I thank the authors for the extensive work made to clarify the point.

3. Thank you for the precisions.

4. This was not quite my question. However, Figure S10 answers it. Thank you.

5. See point 1.

12. This is indeed clearer, thank you. I would, however, change the word “implemented” as the authors do not implement a force but rather exploit them.

14. This remains confusing. Notwithstanding the understandable limitations of experimental resolution, can the authors compare these maps and quantitatively show that they are comparable. Maybe my perception is inaccurate ad result from the different colorbars. More importantly, the underlying question is: does this potential difference bear any consequence for the authors’ argumentation?

Comment regarding (former) P8. There, I feel the authors did not answer my questions. (1) Why could the law not be verified below the threshold for bubble formation using thermal imaging, and (2), why would this law remain true in the presence of a bouncing bubble?

P9: Thus this threshold corresponds to the limits of the imaging capabilities?

[1] Benjamin Dollet, Sander M. van der Meer, Valeria Garbin, Nico de Jong, Detlef Lohse, Michel Versluis, Nonspherical Oscillations of Ultrasound Contrast Agent Microbubbles,

Ultrasound in Medicine & Biology, Volume 34, Issue 9, 2008, Pages 1465-1473
[2] Zeng B, Chong KL, Wang Y, Diddens C, Li X, Detert M, Zandvliet HJW, Lohse D.
Periodic bouncing of a plasmonic bubble in a binary liquid by competing solutal and thermal
Marangoni forces. Proc Natl Acad Sci U S A. 2021 Jun 8;118(23):e2103215118. doi:
10.1073/pnas.2103215118. PMID: 34088844; PMCID: PMC8202017.

Reviewer 1

In the revised manuscript, the authors conducted additional experiments with different cover materials. The underlying mechanism is further verified. They also provided detailed analysis of different kinds of forces experienced by a bouncing bubble. Overall, they properly replied to my comments. Here I only have three minor comments before the manuscript can be accepted.

Thanks for acknowledging that “properly replied to my comments” and “only have three minor comments before the manuscript can be accepted”! These minor comments will be addressed as below.

1. Throughout the main manuscript and supplementary materials, different values of temperature gradient were used. For example, the authors used 10 K/mm on page 4 and 6. On page 18, they used 50 K/mm. In the supplementary materials, they used 30 K/mm (page 18) and 100 K/mm (page 21). Authors should make the value consistent throughout the manuscript. Otherwise, it confuses readers when it comes to different values.

Thanks for the nice comment! In the revised manuscript and supplementary materials, the value of temperature gradient has been corrected to be consistent at 50 K/mm throughout the manuscript.

As shown in Supplementary Fig. 4e, the magnitude of temperature gradient within temperature inversion layer ranges from 20-50 K/mm in experiments for laser power $P=15$ W. Thus, the expression “10 K/mm” on page 4 and 6 has been updated to “50 K/mm”, which is also the chosen magnitude of temperature gradient in the simulation for bouncing bubble. In the supplementary materials, the value 30 K/mm (page 18 in previous version) and 100 K/mm (page 21 in previous version) has been updated to 50 K/mm; the corresponding estimated value of C_1 has been updated to 14.1, instead of 10.

2. Please check if the expression of the upward Marangoni force in Eq. S21 on page 20 of the supplementary materials is correct. Authors used the upward Marangoni force expression corresponding to Eq. S17, which is the case when a bubble completely moves outside TIL. Since H_t is within 0.1 of bubble radius, I assume the bubble is mostly immersed in TIL. Then Eq. S16 other than Eq. S17 is applicable. Can authors clarify this?

Thanks for the nice point! Although temperature inversion layer exists, the layer itself is not fixed during bubble moving in water, as indicated by the oscillation of thickness of temperature inversion layer in Supplementary Fig. 4a.

The expression of the upward Marangoni force depends on not only the location of bubble, but also the motion status of bubble: when bubble rests near the cover, the temperature inversion layer is formed, and the expression of the upward Marangoni

force is $F_m^+ = \Delta\gamma \cdot R = \frac{d\gamma}{dT} \frac{dT^+}{dz} \cdot \max(0, 2R - \delta_{inv}) \cdot \pi R$; when bubble moves in liquid, the temperature inversion layer is disturbed, and the expression of the upward Marangoni force is $F_m^+ = \Delta\gamma \cdot R = \frac{d\gamma}{dT} \frac{dT^+}{dz} \cdot \pi R^2$.

To clarify this point more clearly, some revisions have been made in the section “Mechanism of the bubble bouncing” in main text.

- The expression “depending on the location of the bubble (Fig. 3A, C)” has been revised to “depending on the location and motion status of the bubble (Fig. 3a, c)” on Page 5, line 18.
 - The expression “When the bubble is outside TIL (Fig. 3A)” has been revised to be “When the bubble moves in liquid (Fig. 3a)” on Page 5, line 19.
 - The expression “Once the bubble is mostly immersed in TIL (Fig. 3C)” has been revised to “When the bubble rests near the cover (Fig. 3c)” on Page 5, line 22.
 - The expression “Sketch of ... within (A, B) and outside of (C, D) the TIL, respectively” has been revised to “Sketch of ... when bubble moves in liquid (a, b) and rests near the wall (c, d), respectively” in the caption of Figure 3.
3. In Fig.S9, does the temperature gradient outside TIL need to be zero? If not, please correct it.

Thanks for the nice point! The temperature gradient outside TIL can be positive or zero ($dT^+/dz \geq 0$), although its value is very small. In the revised supplementary (Supplementary Fig. 9), the expression outside TIL “ $dT/dz=0$ ” has been corrected as “ $dT^+/dz \geq 0$ ”, and the temperature gradient within TIL “ $dT/dz < 0$ ” has been corrected as “ $dT^+/dz < 0$ ”. To keep these expressions consistent throughout the manuscript, sketches in Supplementary Fig. 11 and Supplementary Fig. 12 are also revised accordingly.

Reviewer 2

I thank the authors for their detailed answers and the many clarifications they have brought to both the paper and the supplementary materials. They have clarified the major concerns I had. Their analysis and discussions are, now, presented in a clear, refutable and convincing way. I think the work they present is worth being published.

Thanks for acknowledging the improvement of the manuscript, and we appreciate that “the work they present is worth being published”!

I think they should still adjust one explanation regarding point 1 of my previous report, which does not change the following of their analysis. The bouncing phenomenon is observed for the ‘long-time’ regime of the heating of the liquid by the laser. Therefore, equations S3 and S4 are not relevant. They should rather be replaced by the steady, radial heat equation $(1/r)d(r \kappa \quad dT /dr)/dr + A0P \exp(-r^2/r_l^2) = 0$ and its solution, which is also linear in P.

Thanks for the nice point! The analysis of steady temperature for the “long-time” regime is added in Supplementary Discussion 1 “Model for peak temperature evolution” in the revised supplementary materials as suggested.

“For temperature evolution during the long time period, by neglecting convection and considering the thermal diffusion along the r direction, the steady-state axisymmetric heat conduction equation is,

$$\frac{1}{r} \frac{\partial}{\partial r} \left(r \frac{\partial T}{\partial r} \right) + \frac{A_0 P}{\kappa} e^{-r^2/r_l^2} = 0$$

The boundary conditions are as below,

$$\begin{aligned} \left. \frac{dT}{dr} \right|_{r=0} &= 0 \\ T(r_w) &= T_0 \end{aligned}$$

Here the first boundary condition results from the symmetry, and the second condition is a constant temperature for the far field condition, where r_w is the width of water bulk ($r_w= 5$ mm in experiments).

For sufficiently long time period, the diffusion length of heat transfer is much longer than the laser beam, and the laser intensity profile (the term e^{-r^2/r_l^2}) is simplified to a point source of heat. Then, according to Ref. (2), the analytical solution is,

$$\Delta T = T(r) - T_0 = \frac{A_0 P r_w^2}{4\kappa} \left(1 - \frac{r^2}{r_w^2} \right)$$

which reaches a constant temperature for long time period, thus transition regime is observed for intermediate time period as shown in Supplementary Fig. 4b and

Supplementary Fig. 4c.

This also suggests that the authors should simply remove all detailed discussions regarding the short-time (unsteady) regime when the liquid temperature is increasing. This is unnecessary and may bring confusion.

Thanks for the nice comment! Since the short-time (unsteady) regime is responsible for the analysis for bubble dancing when laser is translated horizontally, as shown in Supplementary Discussion 5 “Model for the dancing bubble”. Hence, in the revised supplementary, the discussion for the short-time (unsteady) regime has been kept with some modifications, and the discussion for the long-time (steady) regime has been added as aforementioned.

Minor: the prefactor in (S15) is not correct, it should be 6π for a solid particle and less (depending on surface condition) for a bubble.

Thanks for the nice point! In our experiments, the Reynolds number for bubble motion $Re = \frac{\rho UD}{\mu} \approx 10^2$, where the density of water $\rho = 10^3 \text{ kg/m}^3$, the characteristic velocity for bubble motion $U = 0.1 \text{ m/s}$, the characteristic length for bubble $D = 1 \text{ mm}$, and the viscosity of water $\mu = 1 \text{ mPa} \cdot \text{s}$. According to previous work on the viscous drag force on a spherical bubble [Physics of Fluids 10, 550 (1998)], when Re is large, the viscous contribution $F_v(t) = 12\pi\mu R(t)U$.

In the revised manuscript, the reference is added on page 8 as “the Stokes viscous drag force is raised with v_l ($F_v = -12\pi\mu R v_l$ (31))”. In the revised supplementary materials, the reference is added on page 9 as “the viscous force according to Ref. (5) is:”

Reviewer 3

I thank the authors for carefully considering and addressing my comments. The answers are mostly satisfactory. There are only a few aspects with which I am not fully satisfied. I recommend conditional acceptance, pending the last few revisions.

Thanks for acknowledging “The answers are mostly satisfactory” and recommending “conditional acceptance, pending the last few revisions”, which will be addressed as below.

The authors have written a very interesting discussion on alternative technologies. However, I perceive the discussion as flawed in some respects. Without into the details of specific implementations, both optical and acoustical tweezer have been used to manipulate bubbles. Although I have not used the techniques myself, optical tweezers seem suitable for 3D bubble manipulation without “complex” grating by adapting the laser profile (see [1]). Likewise, SAW is a specific implementation, but acoustic tweezers can also be conceived with transducer arrays, which is a fairly common technology. I do recognize the size-dependent direction of the force as a difficulty.

In short, acoustical and optical tweezer can be used for 3D bubble manipulation. The technique presented by the authors allows mainly 2D manipulation with the constrain of a highly conduction interface in immediate vicinity. The discussion proposed thus need to be revised to account for (1) the fact that the proposed limitation for acoustical tweezers only apply on SAW-based systems (unless shown otherwise), (2) that the limitation on the optical tweezer is thus not justified, and (3) the fact that the discussion does not explicitly explain the added value of the new methods (even if the added value is potentially found for specific cases). I think a clear and solid discussion on the fundamental point is necessary for the broad audience of Nature communications.

[1] Benjamin Dollet, Sander M. van der Meer, Valeria Garbin, Nico de Jong, Detlef Lohse, Michel Versluis, Nonspherical Oscillations of Ultrasound Contrast Agent Microbubbles, *Ultrasound in Medicine & Biology*, Volume 34, Issue 9, 2008, Pages 1465-1473.

Thanks for the nice comment! It is true that acoustic and optical tweezers can be used for 3D bubble manipulation. For optical tweezers, several developed strategies by creating a “dark” trap such as Laguerre-Gaussian beams (see [1]), modulated optical vortices, and high-order Bessel beams have been used for the trapping of low-index particles (like bubbles in liquid medium). But the forces provided by optical tweezers are small, typically on the order of piconewtons, which can only be dominant in the microscopic world [H. Takahira, *JSME Int. J.* 43, 3 (2000)].

Here, in our manuscript, the driving forces originated from the temperature-induced Marangoni effect, are several (or more specific) orders of magnitude larger than optical ones, which can induce larger and longer-range manipulations [A. Miniewicz, *Phys. Chem. Chem. Phys.* 19, 28 (2017), N. A. Ivanova, *Tech. Phys. Lett.* 32, 10 (2006)].

For acoustic tweezers, it can accommodate larger samples [D. Baresch, Phys. Rev. Lett. 116, 2 (2016)]. A recent work indeed reports the 3D trapping of microbubbles by using an acoustic vortex beam which resembles dark optical traps [D. Baresch, PNAS 117, 27 (2020)]. To choose the optical method or the acoustic approach is primarily dependent on the specific application [K. Dholakia, Nat. Rev. Phys. 2, 9 (2020)].

To discuss the contribution or the added value of our method, we think the underlying physical mechanism is worth to be addressed, which is based on the unique temperature inversion layer and leads to the bouncing or even dancing behavior of the underwater bubble..

Hence, in the revised manuscript, we modify the discussion part as below,

“Optical and acoustic tweezers are powerful tools for the non-contact manipulation of bubbles (37, 38). For optical tweezers, the high degree of flexibility and fine spatial resolution is offered, but the achievable trapping force is quite weak on the order of piconewtons, which can only be dominant at the micro-level (39). Acoustic-based traps, which resemble optical traps, can accommodate larger samples with the size on the order of millimeters (40). Although the hybrid trap by combing the two modalities has been implemented, more compact and versatile manipulation designs for multiple functionalities are needed to be further explored (41).”

This are convincing example. Can the authors maybe make use of the half line left in the introduction to explicitly mention the most relevant applications?

Thanks for the good suggestion! We refine the final sentence in the introduction as below:

“consequently shedding light on the bubble-based compositions (such as bubble/droplet capsules or soft robots [4]) for fabrication in materials science, micro-reaction in chemical engineering, wastewater treatment in environmental science, and targeted drug delivery in bioengineering.”

For the uniqueness:

(1) Bouncing of a bubble due to laser irradiation was also observed in [2] (cited by the authors).

[2] Zeng B, Chong KL, Wang Y, Diddens C, Li X, Detert M, Zandvliet HJW, Lohse D. Periodic bouncing of a plasmonic bubble in a binary liquid by competing solutal and thermal Marangoni forces. Proc Natl Acad Sci U S A. 2021 Jun 8;118(23):e2103215118. doi: 10.1073/pnas.2103215118. PMID: 34088844; PMCID: PMC8202017.

(2-4) I agree.

I also concur with the rest of the argumentation. In hindsight, I think it is precisely point (1) that triggered me since the bouncing itself is not new, but the detail physical mechanism is. I would encourage the authors to point out this distinction. Last point with respect to the “obvious superiority”, I have yet to see any proof in that regard

as this is not trivial to me. Maybe a topic for a follow-up investigation.

Thanks for the insightful comment! It is true that the bouncing itself is not new, but the detail physical mechanism is. As claimed in the discussion part, the bouncing bubble via laser irradiation has been observed in [2], but the bounce behavior is identified due to the competition between the solutal and thermal Marangoni forces in the mixed solution of water and ethanol, where the solutal Marangoni force pushes bubble away the substrate, while the thermal Marangoni force sucks the bubble toward the substrate.

In contrast, in our present manuscript, only using of the bidirectional thermal Marangoni forces, the bouncing is achieved in the pure water. Besides, in our present work, the bounce behavior can be coupled with a horizontal translation to achieve 3D manipulation of bubble, which is novel and intriguing here.

The “obvious superiority” refers to the enriched concentration, flexible trajectory, or other aspects, and we will put further investigation on it as a follow-up study.

other comments/concerns:

1. I am confused by the argument. First, the consideration of a rather cylindrical geometry or spherical symmetry depends on the attenuation of the laser as it is absorbed. I do not have the information to know which configuration is appropriate. However, a strictly cylindrical problem requires infinite power (or at least a heating power per unit length) and does not have a steady state. A non-diverging solution can only exist for a finite size source in 3D space. Nevertheless, removing the problematic discussion from the text is fine with me.

Thanks for the comment! We will further investigate the temperature evolution in our future work.

2. I thank the authors for the extensive work made to clarify the point.
3. Thank you for the precisions.
4. This was not quite my question. However, Figure S10 answers it. Thank you.
5. See point 1.

Thanks for the comment! We have refined the discussion on acoustic and optical tweezers in the revised manuscript (see point 1).

12. This is indeed clearer, thank you. I would, however, change the word “implemented” as the authors do not implement a force but rather exploit them.

Thanks for the comment! The word “implemented” has been corrected as “exploited” in the revised manuscript.

14. This remains confusing. Notwithstanding the understandable limitations of experimental resolution, can the authors compare these maps and quantitatively show that they are comparable. Maybe my perception is inaccurate ad result from the

different colorbars. More importantly, the underlying question is: does this potential difference bear any consequence for the authors' argumentation?

Thanks for the nice point! We have quantitatively analyzed the temperature distribution in experiments and simulation (see Fig.2B the blue solid line and Fig.2E blue solid line), and indeed have identified that the magnitude of temperature gradient within and outside the temperature inversion layer is comparable. The mismatch between Fig. 2a and Fig. 2d may arise from the different scalebars and colorbars.

Minor comments:

Comment regarding (former) P8. There, I feel the authors did not answer my questions. (1) Why could the law not be verified below the threshold for bubble formation using thermal imaging, and (2), why would this law remain true in the presence of a bouncing bubble?

Thanks for the nice comment! We have verified the scaling law of $\delta_{inv} \propto P^{-1/4}$ experimentally in the revised Supplementary Discussion 3 "Model for temperature inversion layer", as shown in Supplementary Fig. 6d. This -1/4 scaling law still holds in the presence of a bouncing bubble.

Supplementary Fig. 6 (d) The scaling relation between the thickness of TIL δ_{inv} and laser power P .

P9: Thus this threshold corresponds to the limits of the imaging capabilities?

Thanks for the nice point! Yes, as shown in Fig. 1c, the oscillation amplitude of the bubble center gradually increases with the laser power. Thus, the lower threshold for identifying bubble bouncing in experiments might be limited by the imaging capabilities.